# Controlled assembly of retinal cells on fractal and Euclidean electrodes

**Saba Moslehi** [1,2], **Conor Rowland**[1,2], **Julian H. Smith**[1,2], **William J. Watterson**[1,2], **David Miller**[1,2,3], **Cristopher M. Niell**[4,5], **Benjamín J. Alemán**[1,2,3,6], **Maria-Thereza Perez**[7,8]*, **Richard P. Taylor**[1,2,6]*

1 Physics Department, University of Oregon, Eugene, Oregon, United States of America, 2 Materials Science Institute, University of Oregon, Eugene, Oregon, United States of America, 3 Oregon Center for Optical, Molecular and Quantum Science, University of Oregon, Eugene, Oregon, United States of America, 4 Institute of Neuroscience, University of Oregon, Eugene, Oregon, United States of America, 5 Department of Biology, University of Oregon, Eugene, Oregon, United States of America, 6 Phil and Penny Knight Campus for Accelerating Scientific Impact, University of Oregon, Eugene, Oregon, United States of America, 7 Division of Ophthalmology, Department of Clinical Sciences Lund, Lund University, Lund, Sweden, 8 NanoLund, Lund University, Lund, Sweden

* rpt@uoregon.edu (RPT); maria_thereza.perez@med.lu.se (MTP)

**Data Availability Statement:** All the data and the related code is now available at: https://doi.org/10.6084/m9.figshare.19127894.

**Funding:** RPT, CMN, and BJA: • W. M. Keck Foundation http://www.wmkeck.org/ The funders

## Abstract

Controlled assembly of retinal cells on artificial surfaces is important for fundamental cell research and medical applications. We investigate fractal electrodes with branches of vertically-aligned carbon nanotubes and silicon dioxide gaps between the branches that form repeating patterns spanning from micro- to milli-meters, along with single-scaled Euclidean electrodes. Fluorescence and electron microscopy show neurons adhere in large numbers to branches while glial cells cover the gaps. This ensures neurons will be close to the electrodes' stimulating electric fields in applications. Furthermore, glia won't hinder neuron-branch interactions but will be sufficiently close for neurons to benefit from the glia's life-supporting functions. This cell 'herding' is adjusted using the fractal electrode's dimension and number of repeating levels. We explain how this tuning facilitates substantial glial coverage in the gaps which fuels neural networks with small-world structural characteristics. The large branch-gap interface then allows these networks to connect to the neuron-rich branches.

## Introduction

Experimental fascination with transferring electrical signals to the body predates any sophisticated understanding of the body's wiring components. In 1755, Le Roy sent pulses through a wire wrapped around a blind patient's head which induced perceived flashes of light. This was followed by Galvani's electrophysiological studies of frog muscles [1]. Another century then passed before Ramón y Cajal proposed the neuron doctrine, introducing the modern picture of nervous system wiring as a network of discrete components. Electronic miniaturization now offers surgeons the opportunity to implant devices rather than relying on external wires. Through combined advances in electronics and neuroscience, the prospect of replacing damaged body parts with artificial implants is being transformed from science fiction to science

had no role in study design, data collection and analysis, decision to publish, or preparation of the manuscript. RPT, WJW, and MTP: • The Pufendorf Institute https://www.pi.lu.se/en/pufendorf-ias The funders had no role in study design, data collection and analysis, decision to publish, or preparation of the manuscript. RPT: • The Living Legacy Foundation https://www.thellf.org/ The funders had no role in study design, data collection and analysis, decision to publish, or preparation of the manuscript. • The Ciminelli Foundation The funders had no role in study design, data collection and analysis, decision to publish, or preparation of the manuscript. • University of Oregon https://www.uoregon.edu/ The funders had no role in study design, data collection and analysis, decision to publish, or preparation of the manuscript. MTP: • The Swedish Research Council - # 2016-03757 https://www.vr.se/english.html The funders had no role in study design, data collection and analysis, decision to publish, or preparation of the manuscript. • NanoLund at Lund University https://www.nano.lu.se/ The funders had no role in study design, data collection and analysis, decision to publish, or preparation of the manuscript. • Stiftelsen för Synskadade i f.d. Malmöhus Län https://synskadademalmohus.se/ The funders had no role in study design, data collection and analysis, decision to publish, or preparation of the manuscript. • Crown Princess Margareta's Committee for the Blind The funders had no role in study design, data collection and analysis, decision to publish, or preparation of the manuscript.

**Competing interests:** The authors have declared that no competing interests exist.

fact using intricate electrodes to interface with neurons in e.g., the retina, the ear, the brain, and limbs. Electronic devices have been implanted into human retinas in the hope of restoring vision to patients with degenerative retinal diseases [2–10]. Similarly, over 150,000 deep brain stimulation implant surgeries have been performed targeting neurological disorders such as Parkinson's disease [11], and people with amputated limbs receive interactive prosthetic implants that restore mobility.

Retinal devices have been the focus of broad interdisciplinary research [6, 12–18]. In parallel to medical applications, exploring how retinal cells interact with artificial objects can be used to learn about their fundamental behavior and the extent to which it can be manipulated. In addition to exploring how cells adapt in response to the object's characteristics, it is important to determine the conditions that encourage cells to maintain their natural behavior during interactions. Significantly, implants are frequently referred to as bionic devices in recognition of the importance of bio-inspiration and the need for biocompatibility at the electrode-cell interface. In addition to the electrodes' chemical composition, improvements to their physical properties are crucial and should include favorable mechanical flexibility and material texture. The rigidity of typical electrodes [19] can trigger a reactive response from the retina's glial cells which act as the neurons' life-support system [20, 21]. The resulting glial 'scar' engulfs the implant, increasing the distance to the targeted neurons and degrading the electrode's stimulating power [22–26]. In contrast, materials with a flexibility and surface texture matching those of the biological tissue reduce the gliotic response [23, 27]. The development and maturation of neurons is also promoted when they adhere to textures that mimic their extracellular matrix (ECM) [28].

Materials textured with nano-roughness have the further advantage of increasing the electrode's effective surface area and so their capacity to hold extra surface charge [29], leading to larger stimulating electric fields. These material advantages can be integrated with patterning technologies to modify the electrode's lateral shape, offering the possibility of guiding the neurons and glial cells to different locations within the electrode design. Building on studies demonstrating that neurons accumulate on textured substrates [30–34] and glia on smooth substrates [27, 32, 35–37], gallium phosphide (GaP) nanowires have been used to separate the cells in a co-culture—the texture of the nanowires increased neuron adhesion while glial proliferation occurred simultaneously in neighboring smooth regions [38]. This lateral patterning of surface texture variations is advantageous over chemical steering because the long-term stability of these electrode properties ensures neuron and glial cell survival. The GaP study employed parallel rows of nanowires and smooth regions. The promise of bio-inspiration raises the central question of our study—how do retinal cells assemble on electrodes that mimic the shapes of the networks that neuronal cells normally form?

Many of nature's structures benefit from the favorable functionality that results from fractal branching patterns that repeat at multiple scales [39–41], such as cardiovascular, respiratory [42], and neural systems [43]. Structures with higher fractal dimensions $D$ reduce the size of their repeating patterns at slower rates than those with lower $D$ values. It has been shown that the $D$ value of dendrites growing out from the somas of individual neurons optimizes their connectivity to neighboring neurons [43]. The branching patterns of glial cells can also exhibit fractal characteristics. For both neurons and glia, $D$ has been employed in a diverse range of studies to categorize cell morphologies between different subcategories of the cells [44–53] and also for diagnosing pathological conditions [52–58].

Adopting fractal patterning for electrode designs, experiments [59, 60] and simulations [59, 61, 62] have highlighted their favorable electrical properties for stimulating neurons. In particular, H-Tree electrodes (fractals that repeat an H pattern) positioned above photodiodes have been simulated as artificial photoreceptors within retinal implants. Due to the surface area of

the repeating branches generating a large electrical capacitance, the fractal electrodes stimulated all neighboring neurons while an equivalent square electrode stimulated less than 10% of them [59]. Other proposed technological benefits of fractal electrodes include favorable optical properties (including extraordinary transmission whereby the light transmitted through the gaps in the fractal pattern is greater than that expected from a simple pixel count of gap area) [63–66] and an increase in mechanical flexibility [67].

Critically, these advantageous properties become irrelevant if neurons don't have sufficient proximity to the electrode's stimulating fields due to, for example, poor adhesion or the presence of a glial scar. Whereas fractal texturing has been shown to enhance capacitance [68, 69] and cell growth [28, 60, 70, 71], the employment of fractal branch patterns to selectively direct cells to different regions of an electrode has not been considered: the previous studies of directed cell growth have all focused on single-scale (Euclidean) geometries. Our aim is for neurons to adhere in large numbers on the textured branches of the fractal electrode and for glial cells to primarily cover the smooth surface of the gaps between them. Because of this cell 'herding', the glia would be prevented from hindering neuron-electrode interactions but would be sufficiently close to provide trophic and metabolic support to the nearby neurons. The electrode branches would take on the role of the physical scaffold normally provided by the glial cells when supporting neurons in the retina, ensuring neuron-rich electrodes that maximize stimulation.

To conduct this study, we exploit carbon nanotubes (CNTs) as our electrode material and by doing so extend observations of herding to a new material system. A wealth of previous research has studied interactions of CNTs with many cell types including neurons. Their density, stiffness, and topography can be controlled [72–74], and they can be synthesized on, or transferred to, flexible substrates [75, 76]. Due to a combination of their chemical composition and surface texture, CNTs promote neuronal adhesion and increase the number of processes and their growth [77–79]. Choosing from various topographies [80, 81], we adopt a vertically-aligned CNT (VACNT) approach in which a conducting 'forest' of tangled CNTs is patterned on a smooth silicon dioxide ($SiO_2$) substrate. In addition to providing textured regions with intricate lateral shapes, the high aspect ratios of the VACNTs could aid penetration into neural tissue such as the retina [82]. Building on a broad range of CNT compatibility studies (including *in vitro* [83], rat *in vivo* [84], and human ECG tests [85]), we focus on *in vitro* experiments due to the controlled environment in which fluorescence and electron microscopy can be used to examine cell behavior and cell-electrode interactions as they evolve over periods of up to 17 days *in vitro* (DIV). Previous studies demonstrating that electrically-biased CNT electrodes stimulate neurons effectively [86–88] and even boost their signal transmission [77, 78, 89, 90] indicate that our fundamental studies of cell arrangement have large potential for translation to future electrical measurements and applications.

We fabricate H-tree electrodes (Fig 1) with sizes spanning from a few microns to ~ 4 mm match the individual cells (retinal neuronal and glial cell bodies range from a few μm to ~ 30 μm and glial and dendritic domains can be as large as a few hundred μm) to investigate the individual and cell network interactions with the electrodes. We chose the H-Tree in part because it was the focus of the previous electrical simulations but also because it is an established branched fractal with well-defined scaling properties. Acknowledging that even these simple fractals are inherently complex, we first investigate Euclidean electrodes, consisting of textured rows separated by smooth gaps (Tables 1 and 2), to establish the intrinsic herding characteristics of the VACNT material system. We then investigate the impact of fractal designs with features that start at the scales of the Euclidean designs and repeat at increasingly large sizes. We consider the effect of two central fractal parameters, the number of repeating

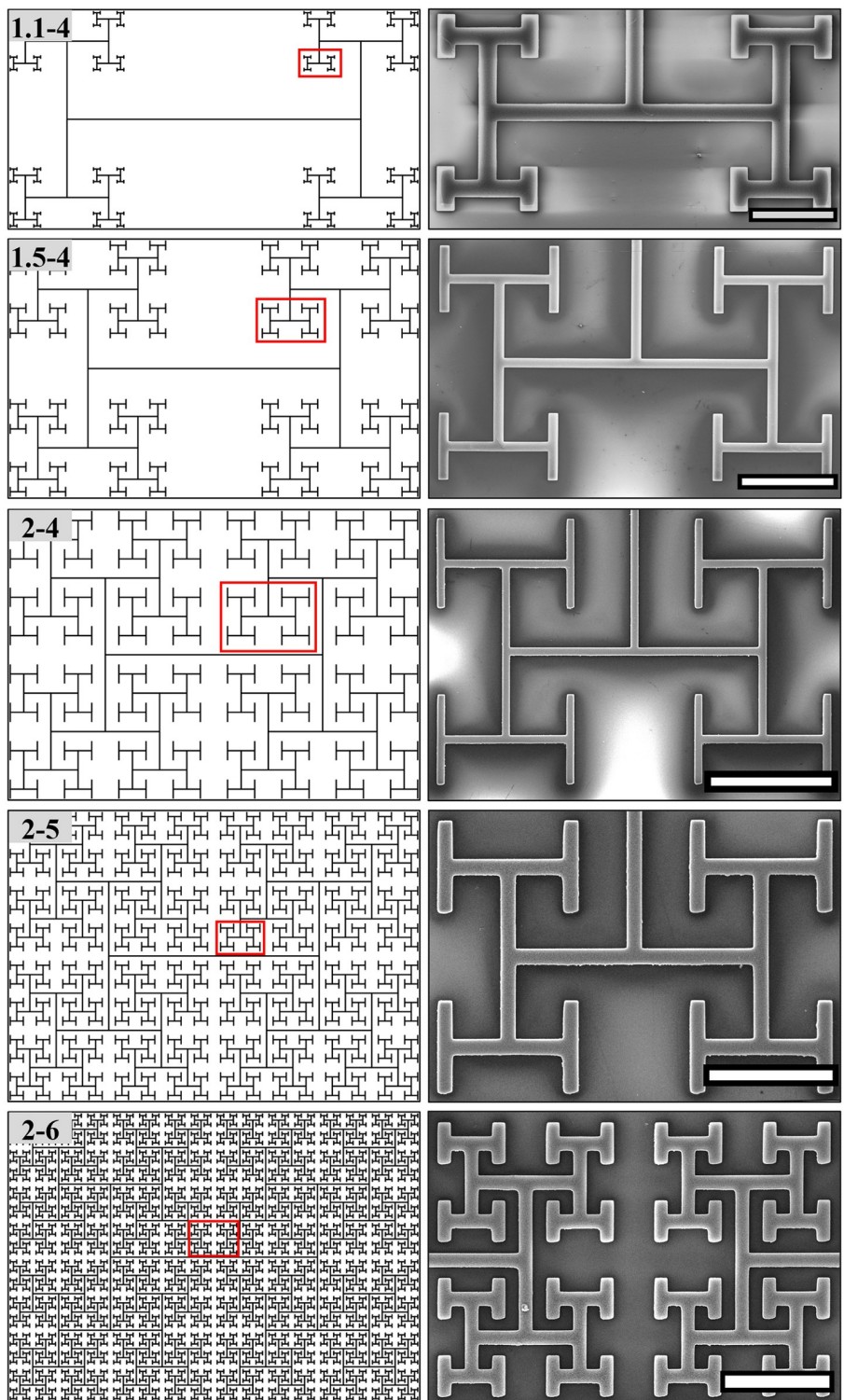

**Fig 1. Schematic and scanning electron microscopy (SEM) images of fractal electrodes used in retinal cell culture experiments with different fractal dimensions *D* and repeating levels *m*.** Left column from top to bottom: ($D = 1.1$ and $m = 4$, labelled as 1.1–4), ($D = 1.5$ and $m = 4$, labelled as 1.5–4), ($D = 2$ and $m = 4$, labelled as 2–4), ($D = 2$ and $m = 5$, labelled as 2–5), ($D = 2$ and $m = 6$, labelled as 2–6). Right column: equivalent SEM image of the marked area in each electrode on the left column. The scale bars are 100, 200, 400, 200 and 200 μm from top to bottom.

**Table 1. Geometric measurements for the Euclidean electrodes.**

| Group | $W_{CNT}$ (µm) | $W_{Si}$ (µm) | $W$ (µm) | $A_{CNT}$ (µm$^2$) | $A_{Si}$ (µm$^2$) | $A_{bounding}$ (µm$^2$) |
|---|---|---|---|---|---|---|
| S100C100 | 100 | 100 | $6 \times 10^3$ | $1.8 \times 10^7$ | $1.7 \times 10^7$ | $3.5 \times 10^7$ |
| S75C100 | 100 | 75 | $6 \times 10^3$ | $2.0 \times 10^7$ | $1.5 \times 10^7$ | $3.5 \times 10^7$ |
| S50C100 | 100 | 50 | $6 \times 10^3$ | $2.4 \times 10^7$ | $1.2 \times 10^7$ | $3.6 \times 10^7$ |
| S25C100 | 100 | 25 | $6 \times 10^3$ | $2.9 \times 10^7$ | $7.0 \times 10^6$ | $3.6 \times 10^7$ |
| S75C75 | 75 | 75 | $6 \times 10^3$ | $1.8 \times 10^7$ | $1.8 \times 10^7$ | $3.6 \times 10^7$ |
| S50C50 | 50 | 50 | $6 \times 10^3$ | $1.8 \times 10^7$ | $1.8 \times 10^7$ | $3.6 \times 10^7$ |
| S25C25 | 25 | 25 | $6 \times 10^3$ | $1.8 \times 10^7$ | $1.8 \times 10^7$ | $3.6 \times 10^7$ |

levels $m$ and the rate of shrinkage between these levels set by $D$, to investigate the effect of incorporating the multiple scales associated with the cell networks.

We combine qualitative with quantitative analysis to demonstrate successful herding for VACNTs patterned with both the Euclidean and fractal geometries. Based on our long-term goal of neuron stimulation and the fact that neuronal processes have a high density of stimulation sites, we focus our quantitative analysis of the neurons on the density of their processes (i.e. total length of the dendrites and axons within a given surface area). Based on their role of promoting neuron homeostasis and survival, we focus the glial analysis on their surface coverage density (referred to hereafter as 'coverage', i.e. the surface area covered by glia normalized to the total area available). We explain the dependence of the herding on $D$ and $m$ in terms of the spatial freedom provided to the glia by the multi-scaled, interconnected gaps and their close proximity to the neuron-rich electrodes. The latter is accentuated by the large interface between the glial-rich gaps and neuron-rich electrodes, generated by the repeating patterns and the resulting interpenetrating, tortuous character of the long fractal branches. In addition to addressing a fundamental question—how do retinal cells respond to fractal electrodes with multi-scaled patterns that span those of the cell networks—our findings are potentially important for future applications involving neuron stimulation.

## Results

### Qualitative observations of herding

We first made qualitative observations for the Euclidean row electrodes using fluorescence and electron microscopy at 17 DIV to establish the basic herding properties of the retinal cells. Large numbers of glial cells were observed in the gaps, but these were confined by the electrodes and never traversed them (Fig 2). Individual glia rarely attached to the electrodes (Fig 2b and 2f) and when doing so typically exhibited a more branched morphology (Fig 2e). In contrast, neuronal processes grew on both the gap and electrode surfaces, although they were

**Table 2. Geometric measurements for the fractal electrodes.**

| Group | $W_{CNT}$ (µm) | $W_{Si\text{-}min}$ (µm) | $W_{Si\text{-}max}$ (µm) | $W$ (µm) | $A_{CNT}$ (µm$^2$) | $A_{Si}$ (µm$^2$) | $A_{bounding}$ (µm$^2$) |
|---|---|---|---|---|---|---|---|
| 1.1–4 | 20 | 56.4 | $4.6 \times 10^3$ | $6.0 \times 10^3$ | $8 \times 10^5$ | $1.8 \times 10^7$ | $1.9 \times 10^7$ |
| 1.5–4 | 20 | 101.0 | $3.7 \times 10^3$ | $6.0 \times 10^3$ | $1.5 \times 10^6$ | $2.1 \times 10^7$ | $2.3 \times 10^7$ |
| 2–4 | 20 | 134.3 | $3.3 \times 10^3$ | $6.1 \times 10^3$ | $2.3 \times 10^6$ | $2.4 \times 10^7$ | $2.7 \times 10^7$ |
| 2–5 | 20 | 61.2 | $3.2 \times 10^3$ | $6.3 \times 10^3$ | $4.6 \times 10^6$ | $2.3 \times 10^7$ | $2.8 \times 10^7$ |
| 2–6 | 20 | 25.0 | $3.2 \times 10^3$ | $6.3 \times 10^3$ | $8.8 \times 10^6$ | $1.8 \times 10^7$ | $2.8 \times 10^7$ |

Note that the measurements for the 2–4 fractals are the averages between the two designs.

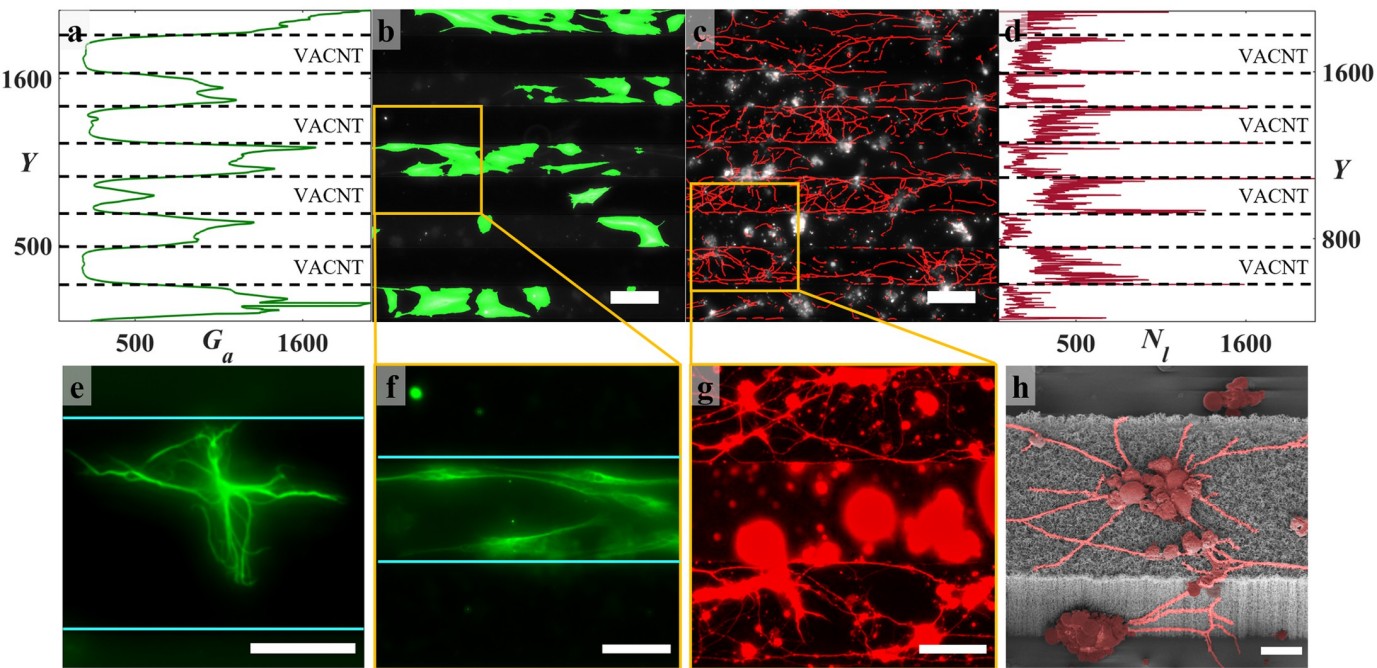

**Fig 2. Neuronal and glial behaviors for Euclidean electrodes imaged at 17 DIV.** (a) Sum of glial coverage areas shown in panel (b) (measured in pixels with a pixel width of 0.32 μm), revealing peaks within the $SiO_2$ gaps. (b) Representative fluorescence image of GFAP labelled glial cells of a S75C75 electrode superimposed on the regions of glial coverage identified by the algorithm (green). (c) Representative fluorescence image of β-Tubulin III labeled neurons of the same region in (b) superimposed on the neuronal processes identified by the algorithm (red). (d) Sum of process lengths (in pixels) shown in panel (c), revealing peaks coinciding with the electrode edges. (e) Representative fluorescence image of a GFAP labelled glial cell on the VACNT top surface of a S75C75 electrode. (f) Zoom-in representative fluorescence image of GFAP labeled glial cells of the area marked in (b). (g) Zoom-in representative fluorescence image of β-Tubulin III labeled neurons of the area marked in (c). (h) SEM image of a S50C50 Euclidean electrode taken at 40˚ tilt showing neuron clusters and connecting processes (false-colored) adhering to the top surface and sidewalls of the electrode (7 DIV). The dotted black lines in (a) and (b) and the cyan lines in (e) and (f) locate the edges of the VACNT rows. Scale bars are 100 μm in (b) and (c), 50 μm in (e), (f), and (g), and 10 μm in (h).

considerably longer and formed more complex networks on the electrodes. For both the gaps and electrodes, neuronal somas were seen to cluster together and the relatively simple networks in the gaps featured fewer but larger clusters (Fig 2c and 2g). The neuronal processes followed the top and the bottom edges of the electrodes upon reaching them and were able to climb up or down the sidewalls to connect cell clusters that existed on both surfaces (Fig 2c, 2g and 2h). Processes occasionally also bridged gaps, connecting the edges of neighboring electrodes (S1a Fig). These effects were analyzed using the neuronal process length and the glial area algorithms (Methods). The glial coverage area $G_a$ and neuronal process lengths $N_l$ were summed along the rows (i.e. in the X direction) to assess the peak glia and neuron locations in the Y direction perpendicular to the rows (Fig 2a and 2d). Whereas $G_a$ peaked within the gaps, $N_l$ was largest on the electrodes and peaked at their edges.

Fig 3 summarizes the retinal cell responses to the fractal electrodes imaged at 17 DIV. One notable characteristic of their multi-scaled geometry is the frequent change in branch direction. Although glia rarely adhered to the electrodes, they elongated themselves along the branches and were not restricted by their 90˚ turns (Fig 3a). Glial cells were observed in the gaps of all fractal electrodes by 17 DIV, even for the most restricted gap connections (See Fig 3b for the 2–6 fractals; we note that fractals with higher *m* were excluded from our study because the smallest repeating levels would have closed to form disconnected gaps). We found that neurons readily grew processes on the electrodes, forming networks that followed their edges and made 90˚ turns at branch junctions (Fig 3e, 3f and 3k).

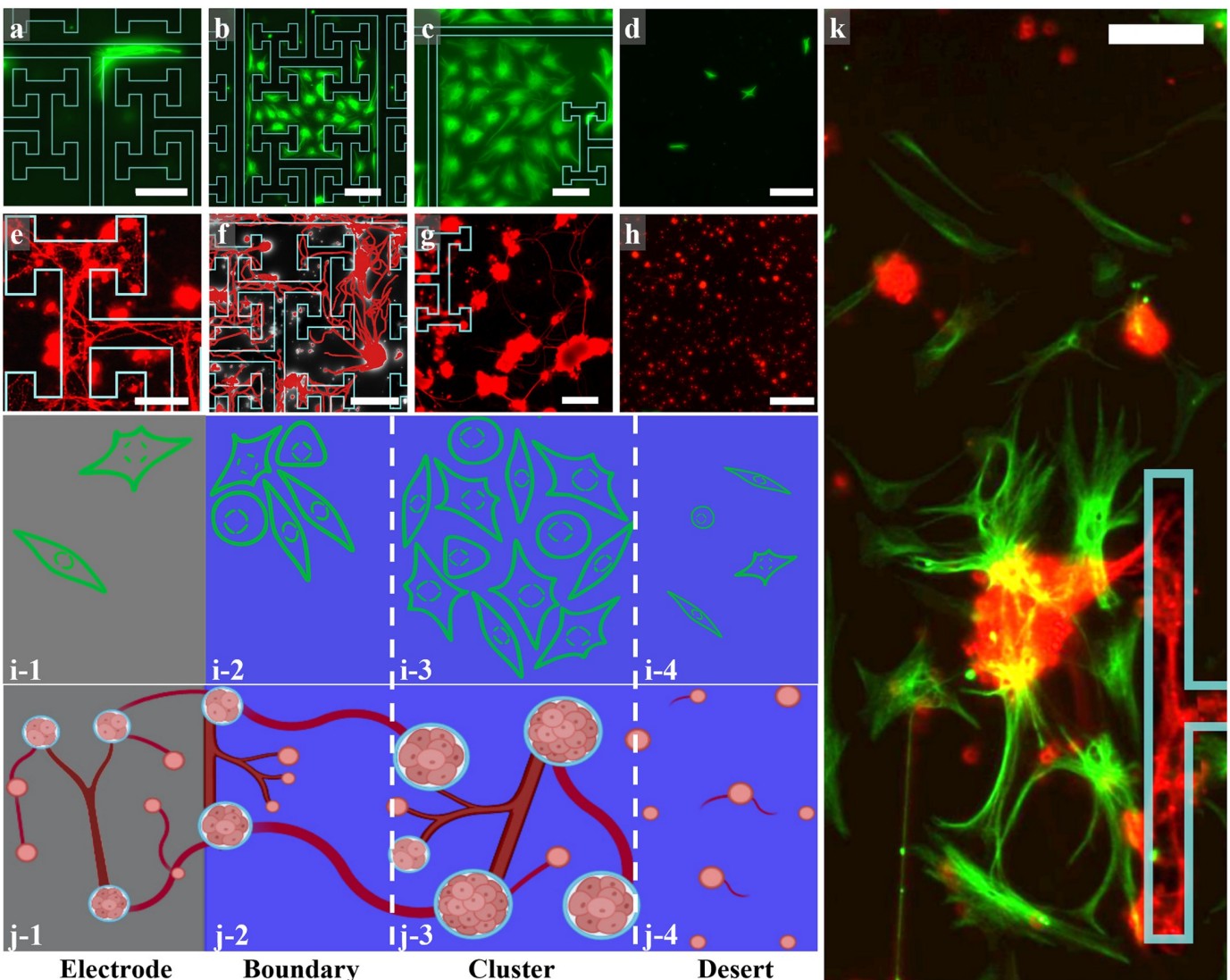

**Fig 3. Examples of fluorescence images of retinal cells interacting with the fractal electrodes at 17 DIV (green = GFAP labelled glia; red = β-tubulin III labelled neurons).** (a) The rare occurrence of glia following the 90˚ turn of a 2–6 electrode branch. (b) Glial coverage in the gap of a 2–6 electrode. (c) Glial coverage in the gap of a 1.1–4 electrode close to its branches. (d) Individual glia in a desert region away from the branches of a 1.1–4 electrode. (e) Neurons and their processes on a 2–6 electrode's branches. (f) Neuron clusters and processes in a boundary region interacting with the neurons on the nearby branches of a 2–6 electrode. Neuronal processes were semi-automatically traced using the Fiji simple neurite tracer and were false-colored. (g) Neuron clusters and processes forming a cluster neural network in the gaps of a 1.1–4 electrode. (h) individual neurons in a desert region of a 1.1–4 electrode far from the branches. (i) and (j) Schematic of the glial and neural network regions. (i-1) and (j-1) show the electrode with few glial cells and multiple processes connecting individual neurons and small to medium-sized clusters. (i-2) and (j-2) show the 'boundary' region featuring small to medium glial coverage regions and clusters connecting to each other and to neurons on the electrodes using multiple processes. (i-3) and (j-3) show the 'small-world' region featuring larger glial coverage and clusters with bundles of processes connecting them. (i-4) and (j-4) show the 'desert' region furthest from electrodes featuring very few glial cells, mostly individual neurons and very few processes. (k) Merged fluorescence image of glia and neurons on a 2–4 electrode showing all the different regions. Scale bars on (a), (b), (c), (f), and (g) are 100 μm, on (d) and (h) are 200 μm, and on (e) and (k) are 50 μm. The electrode edges are highlighted in cyan in (a), (b), (c), (e), (f), (g) and (k). Schematic panels were created in BioRender.

To facilitate more detailed observations, we categorized the gaps into three regions based on cell behavior. Fig 3 shows columns of example images of the neurons and glia, along with schematic representations immediately below these images: the electrode region (Fig 3a, 3e, 3i-1, and 3j-1), the 'boundary' region (Fig 3b, 3f, 3i-2 and 3j-2), the 'cluster' region (Fig 3c, 3g, 3i-

3 and 3j-3), and the 'desert' region (Fig 3d, 3h, 3i-4 and 3j-4). Furthest away from the electrodes were the desert regions, which featured a few individual neurons and small clusters with weak processes, along with a scattering of glial cells. Nearer to the electrodes, neurons aggregated into larger clusters physically connected to each other by bundles of processes and accompanied by significant numbers of glia. These are labelled as the cluster regions—in recognition of these typically larger clusters relative to those in the other regions. Many of these networks were connected to neurons on the electrodes via the boundary regions, which formed in some places along the electrode-gap interface. These boundary regions were composed of small to medium-sized clusters and accompanied by occasional glial coverage. Fig 3k captures these various behaviors in one wide field of view (FOV).

The desert regions were most prevalent for the 1.1–4 electrodes. Their size diminished with increasing $D$ and $m$ until they vanished completely for the 2–5 and 2–6 fractals. In contrast, the contributions of the boundary regions increased with $D$ and $m$, with the 2–6 fractal displaying the most processes connecting from the gaps to the electrodes (Fig 3f). The cluster regions were prevalent for the $D = 1.5$–4 electrodes. Based on this $D$ and $m$ dependence, the sizes of the regions varied between the different electrodes. We will return to the importance of these regional behaviors along with their $D$ and $m$ dependence after we have quantified the herding behavior of the various electrodes.

Having identified these three regions for the fractal gaps, we revisit the Euclidean gaps. These tended to be dominated by boundary regions with an absence of deserts. Although less prevalent than for the fractals, cluster regions were apparent and their time evolution is shown in Fig 4a–4c. These show three regions containing glial cells on both the electrode and gap surfaces at the 3, 7, and 17 DIV. Notably, through cell division and growth, the glia have started to cover increasingly larger areas in the gaps by 17 DIV. Fig 4d–4f show different regions from the same electrodes as a, b, and c, now including the neuronal behavior. Whereas the neuronal processes have grown from 3 to 7 DIV to connect the clusters, 17 DIV reveals fewer but larger clusters connected by bundles of processes, as shown in Fig 4g. Visual inspection reveals that this signature of network formation was mildest on the electrode surfaces when compared to the gaps.

## Quantification of herding

The time evolution for the Euclidean group is quantified in Fig 4h–4k. At each DIV, all the Euclidean electrodes were combined regardless of their sizes (see Methods for the widths of the electrode $W_{CNT}$ and the gap $W_{Si}$). This was justified since performing statistical tests (Methods) revealed no significance in $G_{Si}$, $G_{CNT}$, and $N_{CNT}$ between any pairs. As for $N_{Si}$, no significances were detected with the exception of between $W_{Si} = 25$ and 100 μm at 3 and 7 DIV ($p = 0.013$ and $p = 0.006$, respectively), as well as between $W_{Si} = 50$ and 100 μm at 17 DIV ($p = 0.028$). Consistent with the qualitative observations, $G_{CNT}$ was an order of magnitude smaller than $G_{Si}$ and both increased with time. In contrast, $N_{CNT}$ and $N_{Si}$ exhibited a peak at 7 DIV. Statistical comparisons between all DIV pairs revealed the following results: $G_{Si}$ was significantly lower at 3 DIV than at 7 and 17 DIV ($p \leq 0.001$ and $p \leq 0.0001$, respectively) and was significantly lower at 7 DIV than at 17 DIV ($p \leq 0.001$). $G_{CNT}$ was significantly lower at 3 DIV than at 7 and 17 DIV ($p = 0.033$ and $p \leq 0.0001$, respectively). $N_{Si}$ and $N_{CNT}$ were significantly lower at 3 and 17 DIV than at 7 DIV ($p \leq 0.001$ for all DIV pairs).

Fig 5 summarizes the glial and neuronal behavior in the $SiO_2$ gaps and on the VACNT surfaces of the Euclidean and fractal electrodes at 17 DIV with respect to their effective feature sizes and geometries. Fig 5a and 5c show the relationship of $G_{Si}$ and $N_{Si}$ with $W_{Si}$ for the Euclidean electrodes at 17 DIV. For these plots, Euclidean electrodes with identical $W_{Si}$ but

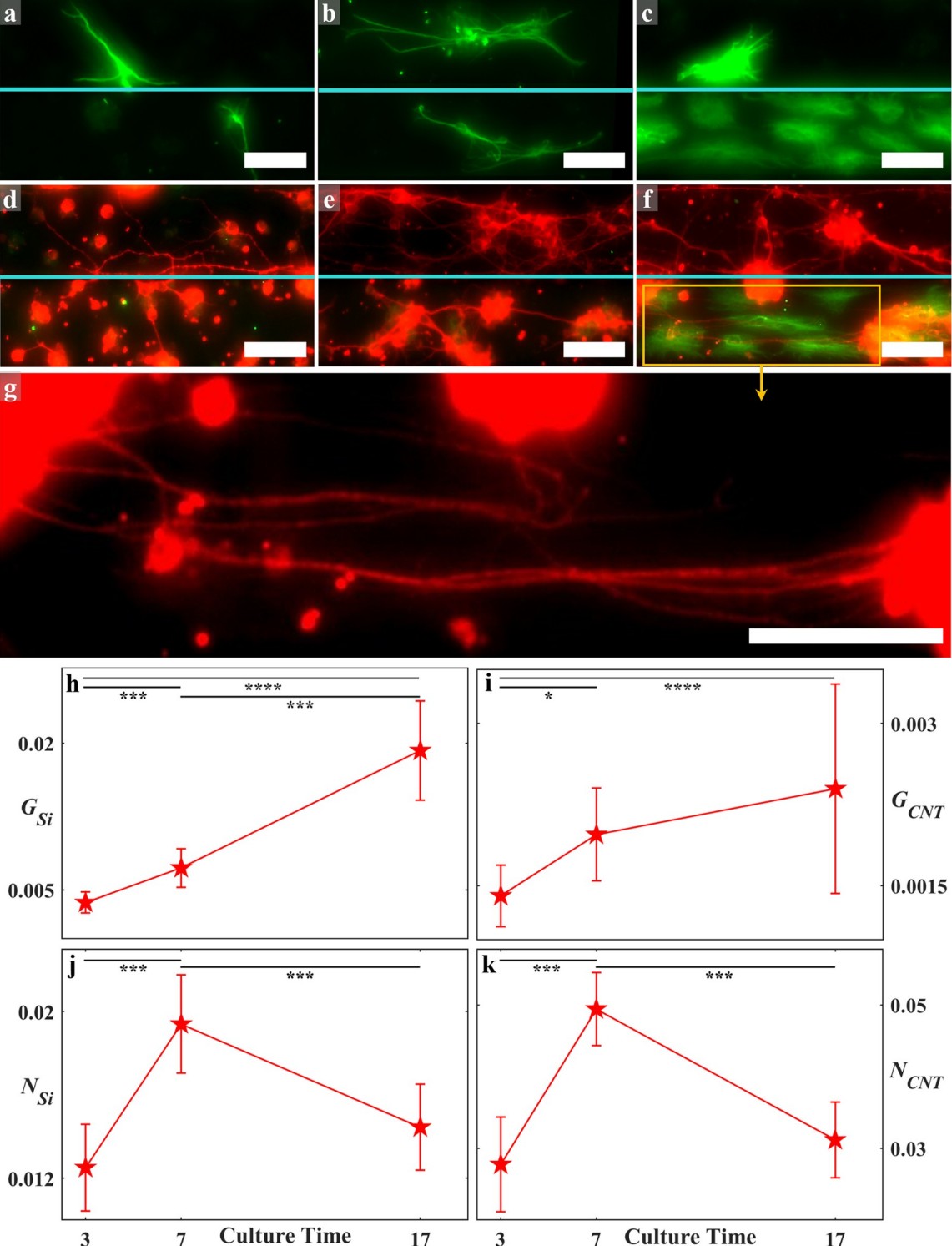

**Fig 4. Examples of fluorescence images of retinal cells interacting with the Euclidean electrodes at all culture times (green = GFAP labelled glia; red = β-tubulin III labelled neurons).** (a, b, c) GFAP labelled glial cells (green) on the VACNT and SiO₂ gaps of S75C75 electrodes at 3 DIV (a), 7 DIV (b) and 17 DIV (c). (d, e, f) Merged fluorescence images of neural networks showing GFAP labelled glia (green) and β-tubulin III labelled neurons (red) on different regions of the same electrodes shown in (a), (b), and (c). Panel (g) is a zoom-in on the region marked in (f) with the green channel removed in order to clearly highlight neuronal processes bundling in the SiO₂ gap. Scale bars are 50 μm in (a) through (g). The cyan lines mark the edge between the VACNT electrode (top half) and SiO₂ gap

(bottom half) in (a) through to (f). (h, i, j, k) Time evolution of $G_{Si}$, $G_{CNT}$, $N_{Si}$, and $N_{CNT}$ for all Euclidean electrodes averaged at each culture time. The glial cells follow a gradual increase in surface coverage across the culture time while the neuronal processes show a peak at 7 DIV (Table 3 shows the number of analyzed electrodes at each culture time). The error bars correspond to the 95% confidence intervals. Stars in (h), (i), (j), and (k) indicate the degree of significance: $^{*}$ denotes $p \leq 0.05$, $^{***}$ denotes $p \leq 0.001$, and $^{****}$ denotes $p \leq 0.0001$.

different $W_{CNT}$ were combined (e.g., S25C25 was combined with S25C100 and so on) since statistical tests showed no significant differences between same $W_{Si}$ subgroups, indicating that $W_{CNT}$ did not significantly impact glial and neuronal growth in the gaps. In Fig 5a, $G_{Si}$ consistently increased with $W_{Si}$ up to 75 μm and then decreased for $W_{Si} = 100$ μm (in agreement with qualitative observations of smaller glial coverage in the S100C100 $SiO_2$ gaps). However, a statistical test showed no significant differences in $G_{Si}$ between any pairs with different $W_{Si}$. Fig 5c shows a gradual decrease in $N_{Si}$ with increasing $W_{Si}$, with the $W_{Si} = 50$ and 100 μm groups being significantly different to each other. Fig 5b and 5d show the relationship of $G_{CNT}$ and $N_{CNT}$ with $W_{CNT}$ for the Euclidean electrodes at 17 DIV. For these plots, the Euclidean electrodes with identical $W_{CNT}$ but different $W_{Si}$ (i.e. all electrodes with $W_{CNT} = 100$ μm) were combined since statistical tests showed no significant differences between any pairs with the same $W_{CNT}$, indicating that $W_{Si}$ did not significantly impact glial and neuronal growth on the VACNT surfaces. No increasing or decreasing trends were observed for $G_{CNT}$ or $N_{CNT}$ and no significant differences were detected between any pairs with different $W_{CNT}$.

In addition, the Euclidean electrodes were compared with regard to their $SiO_2$ to VACNT surface area ratio and its effect on glial and neuronal growth at 17 DIV (S2 Fig). Results of statistical tests showed no significant difference between any pairs from the Euclidean subgroups for any of the neuronal or glial measurements despite the ratio varying by up to a factor of 4 between various electrodes. This showed that there was no measurable toxicity impact of the VACNTs on either the neurons or the glial cells up to 17 DIV for these area ratios.

Fig 5e–5h summarizes the glial and neuronal behavior on both surfaces of the fractal electrodes as a function of $D$ and $m$. $G_{Si}$ peaked for the 1.5–4 fractal, although statistical tests revealed a significant difference only between the 2–5 and 2–6 fractals (p = 0.036). $G_{CNT}$ was more than an order of magnitude smaller than $G_{Si}$ and was almost constant across all fractals, with the 1.1–4 fractal having the lowest value. Statistical tests showed no significant differences in $G_{CNT}$ between any pairs from the fractal subgroups. $N_{Si}$ and $N_{CNT}$ gradually increased with $D$ but not with $m$. Statistical comparisons revealed no significant differences in $N_{CNT}$ between any pairs from the fractal subgroups. As for $N_{Si}$, the following fractal subgroups were significantly different: 1.1–4 vs 2–5 (p $\leq$ 0.001), 1.1–4 vs 2–6 (p = 0.033), and 1.5–4 vs 2–5 (p = 0.044) (S3 Fig).

In terms of herding, when we grouped all of the 17 DIV Euclidean electrodes together, we found that $N_{CNT}$ was significantly higher than $N_{Si}$ (p $\leq$ 0.0001) and that $G_{Si}$ was significantly higher than $G_{CNT}$ (p $\leq$ 0.0001) (S4a and S4b Fig). We found exactly the same results when we grouped the fractal electrodes together, demonstrating the herding power of the VACNT-$SiO_2$ material system for both electrode geometries (S4c and S4d Fig). This also held for comparisons within each of the individual subgroups (Tables 1 and 2) with the exception of the neuron behavior for the S50C50 and S100C100 subgroups.

To further quantify neuron herding power, we introduced $N$ as the ratio of the total process length on the electrode ($N_{CNT}$) to that on the combination of the electrode and gap surfaces (i.e. $N_{CNT} + N_{Si}$), where $N_{CNT}$ and $N_{Si}$ have been normalized relative to the surface areas of the electrode and $SiO_2$, respectively (Methods). Similarly, we introduce $G$ as the ratio of the glial coverage area in the gaps ($G_{Si}$) to that in the electrode and gap surfaces combined (i.e. $G_{CNT} +$

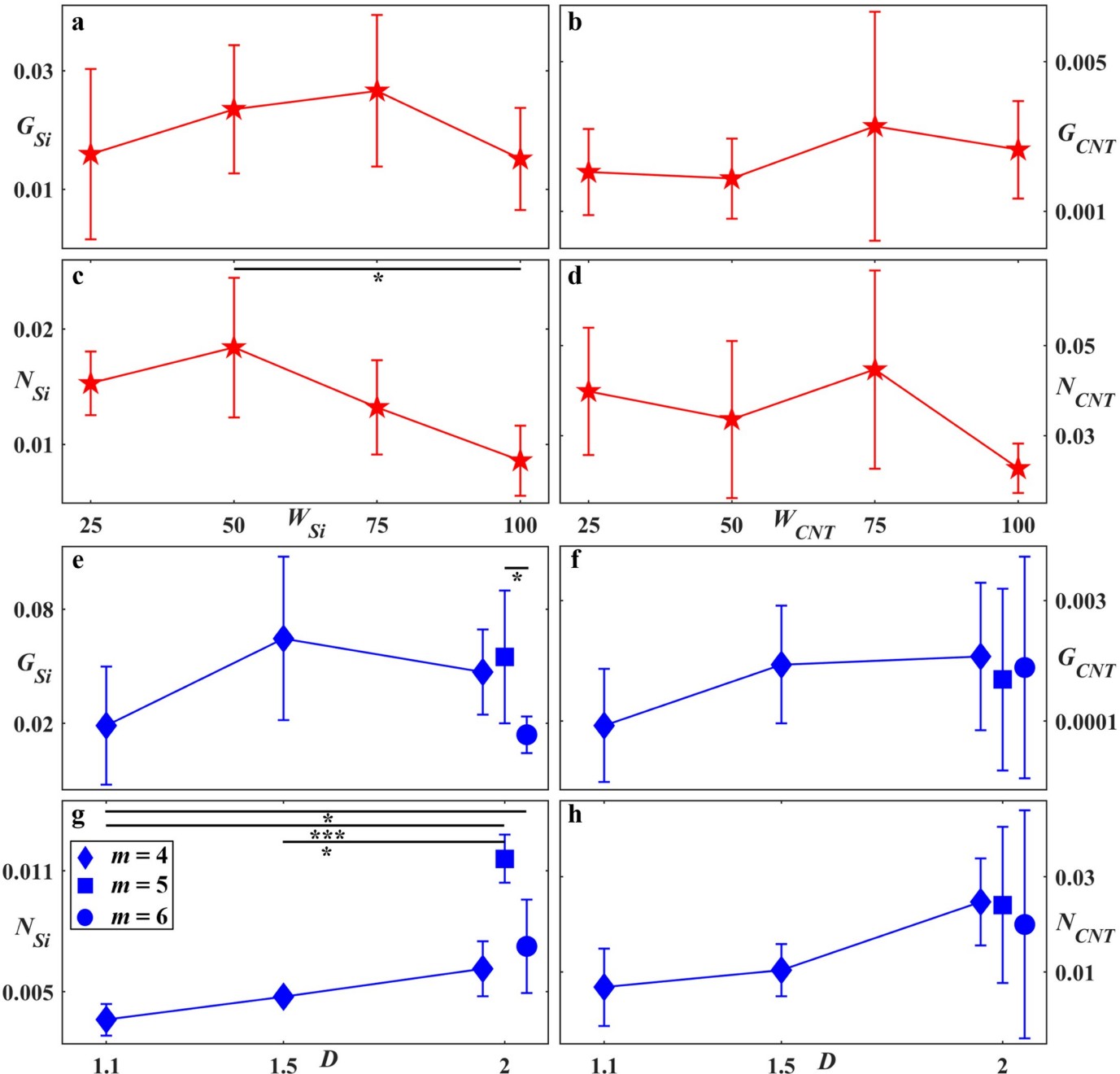

**Fig 5. Glial and neuronal behavior for Euclidean and fractal electrodes at 17 DIV.** (a) $G_{Si}$ median change with $W_{Si}$, (b) $G_{CNT}$ median change with $W_{CNT}$, (c) $N_{Si}$ median change with $W_{Si}$, (d) $N_{CNT}$ median change with $W_{CNT}$. No significance was detected between any Euclidean pairs in panels (a), (b), and (d). In panel (c), significance was detected between $W_{Si} = 50$ and $W_{Si} = 100$ μm (p = 0.018). (e), (f), (g), and (h) show $G_{Si}$, $G_{CNT}$, $N_{Si}$, and $N_{CNT}$ median trend with $D$ and $m$. The 2–4 and 2–6 fractal datapoint are slightly shifted from $D = 2$ for clarity. The error bars correspond to the 95% confidence intervals. Stars in (c), (e), and (g) indicate the degree of significance: $^{*}$ denotes p ≤ 0.05, and $^{***}$ denotes p ≤ 0.001.

$G_{Si}$), where $G_{CNT}$ and $G_{Si}$ have also been normalized to the surface areas of the electrode and $SiO_2$, respectively (Methods). Adopting these measures, $N$ and $G$ powers greater than 0.5 indicate successful guiding of neuronal processes and glial cells to the desired VACNT and $SiO_2$ surfaces, respectively. For quantification of herding, we grouped the electrodes into the

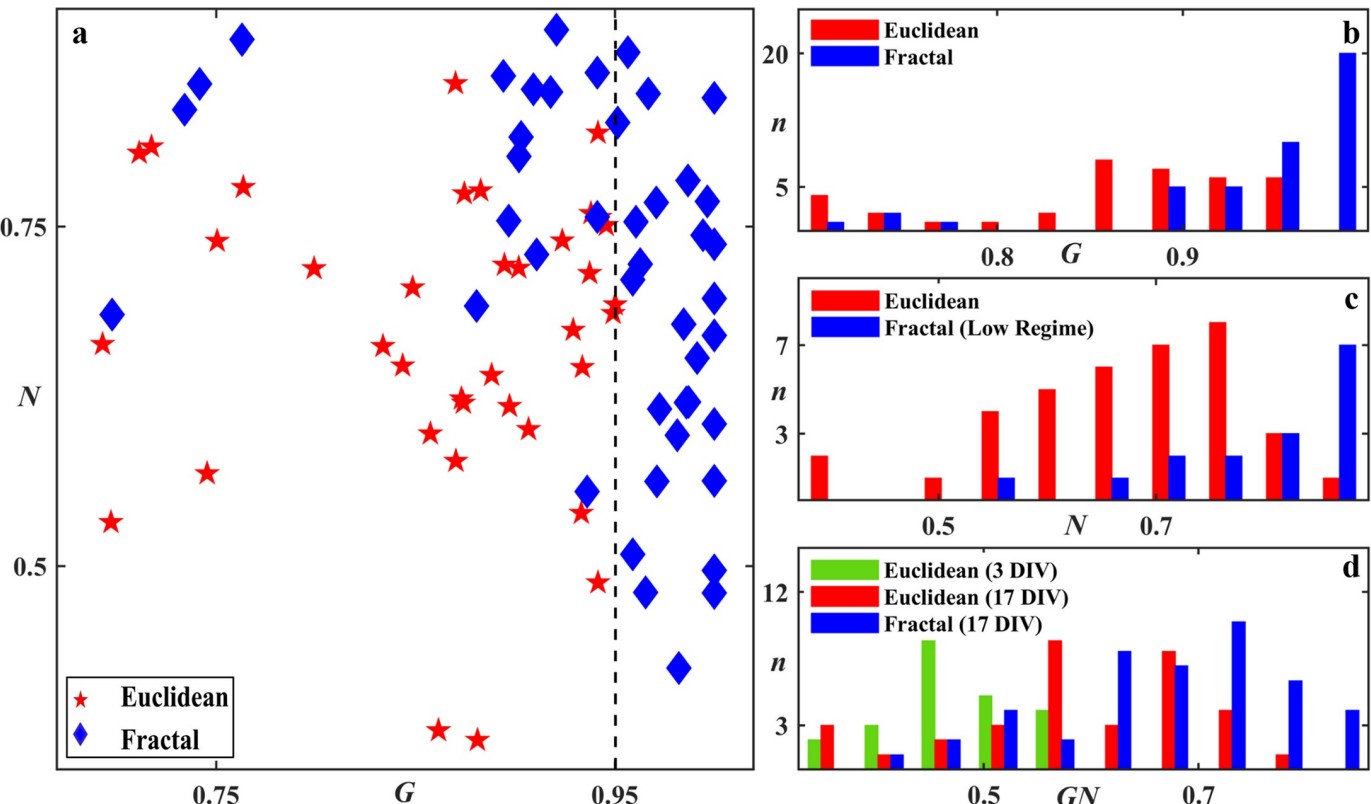

**Fig 6. Quantification of herding.** (a) Scatterplot of $N$ (neuron herding) vs $G$ (glial herding) at 17 DIV, for 38 Euclidean and 44 fractal electrodes where each data point represents one electrode (we display $0.5 < G \leq 1$ for clarity but note that the one Euclidean electrode with $G < 0.5$ not shown here was included in the analysis). The dashed line marks the threshold value in $G$ that no Euclidean electrode surpassed. (b) Histogram of the number of electrodes $n$ with a given $G$ for 17 DIV Euclidean and fractal electrodes. (c) Histogram of the number of electrodes $n$ with a given $N$ for 17 DIV Euclidean and low regime fractal electrodes. (d) Histogram of the number of electrodes $n$ with a given $GN$ for 3 and 17 DIV Euclidean plus 17 DIV fractal electrodes. Euclidean data for 7 DIV is not shown for clarity but follows the observed trend.

Euclidean and fractal groups to provide an overall view of $N$ and $G$ powers. We will return to the individual parameters, $N_{CNT}$, $N_{Si}$, $G_{CNT}$ and $G_{Si}$, in the Analysis Section when investigating their dependences on the various electrode parameters (some of which vary across the Euclidean and fractal groups).

Fig 6a shows a scatterplot of $N$ vs $G$ for the Euclidean and fractal electrodes measured at 17 DIV. The dashed black line represents a threshold $G_T$ in glial herding at $G \sim 0.95$, beyond which no Euclidean electrodes were observed. The fractal electrodes, on the other hand, were not limited by this threshold and achieved significantly higher $G$ values than the Euclidean electrodes ($p \leq 0.0001$. Fig 6b). Based on $G_T$, the fractal electrodes were divided into two regimes in Fig 6a: low ($G \leq G_T$) and high ($G > G_T$) regimes. We note the following overall observations for Fig 6a: 1) almost all electrodes (90% of Euclidean and 95% of fractal electrodes) were successful at herding (i.e. $G > 0.5$ and $N > 0.5$), highlighting the favorable material qualities of the VACNTs for herding, 2) in the low regime, fractal electrodes achieved significantly higher neuron herding values than the Euclidean electrodes ($p \leq 0.001$; Fig 6c), 3) in the high regime, the enhanced neuron herding collapsed such that the Euclidean and high regime fractal electrodes shared the same approximate range of $N$ values.

We also introduce the multiplication parameter $GN$ to quantify the combined herding power. To conduct a statistical analysis, we combined the electrodes into Euclidean and fractal

groups. The Euclidean electrodes exhibited significantly lower *GN* performance compared to the fractals (p ≤ 0.0001, p ≤ 0.0001, and p = 0.004 for 3, 7, and 17 DIV Euclidean versus 17 DIV fractals, respectively). We plot the histogram of the number of electrodes *n* with a given *GN* for Euclidean (3 and 17 DIV) and fractal (17 DIV) electrodes in Fig 6d. This indicates that not only did time evolution increase the combined herding power but also that this power was amplified for the fractal electrodes. S1 Table summarizes the median values of the herding parameters ($G_{Si}$, $G_{CNT}$, $N_{Si}$, $N_{CNT}$, *N*, *G* and *GN*) for all of the various Euclidean and fractal electrode types at each DIV.

## Discussion

### Basic herding behavior

We begin by reviewing the herding properties of the retinal cells on the Euclidean electrodes. The large neuronal process lengths observed on the electrodes is consistent with previous results [77] and can be explained in part by the VACNTs' favorable surface texture: their nano-scale roughness (S1b and S1c Fig) has been proposed to mimic some of the ECM properties [91, 92] and to enhance neurite outgrowth and elongation if the roughness variation matches the process diameter [93]. VACNT flexibility also likely plays a role since neurons are known to readily adhere to and grow processes on softer substrates [94, 95]. Although it has been suggested that CNT functionalization is necessary for biocompatibility, our observations support previous results demonstrating that appropriate degrees of texture and flexibility of pristine VACNTs are sufficient for neural network survival [83], which in turn favors recording and stimulation [96]. The increased process density at our electrode edges is also consistent with previous work showing that neurons respond to topographical cues [38, 97, 98] for a range of feature sizes [99–101] and in particular that they align with the direction of minimum surface curvature, so favoring paths that minimize process bending [102]. In our case, when processes reached the electrode's top surface edge, they were more likely to follow the edge rather than take the 90° turn to grow down the sidewall. The strength of this physical cue was highlighted by its frequent observation even for edges bordering on gaps featuring no glia (Fig 2c and 2g).

In contrast to inducing favorable neuronal responses, the VACNTs severely dampened glial coverage, consistent with experiments showing that CNT-coated brain implants reduce glial responses [81]. Previous experiments aimed at demonstrating the dampening impact of nano-scale features showed that carbon nanofibers limited *in vitro* glial functions [27] and other experiments identified that softer substrates weaken the surface interactions necessary for glial proliferation [95]. This proliferation dependence on hard, smooth surfaces likely explains the observed lack of glial coverage on our VACNT electrodes even at 17 DIV, while extensive coverage was observed in the flat gaps. These results are also in accordance with the previous observations using the same cell system on rows of vertical GaP nanowires separated by areas of flat GaP, in which glial cells were seen to occupy the flat areas while neuronal processes were seen in association with the nanowires [38]. In particular, it is informative to compare the GaP nanowire herding to our VACNT rows with a comparable width (S100C100 Euclidean electrodes in S1 Table). Although the GaP study employed a different method for quantifying *G* and *N* and did so over smaller FOVs, their values are comparable to ours.

### Network formation

To understand the impact on herding of adding the increasingly large fractal branches and gaps to our electrode designs, we need to examine neural network formation in greater detail. As anchorage-dependent cells, neurons rely on surface adhesion for their development,

migration, and ultimate survival. In our system, the neurons tend to aggregate into clusters and gradually establish structural characteristics reminiscent of small-world networks—a class of network so named because each node is connected to all other nodes through a small number of connecting neuronal processes, a configuration that has been proposed to maximize efficiencies such as signal transmission [103, 104]. *In vitro* studies of small-world networks on flat substrate surfaces [105–107] have shown that, once seeded, neurons extend their processes in search of neighboring cells and reach a maximal complexity state featuring a large number of nodes (individual cells and clusters) and links (neuronal processes) between them. Lacking organizational characteristics such as self-avoidance, the network then starts to optimize as it shifts dominance from mostly neuron-substrate to also neuron-neuron interaction forces [108]. The number of nodes then decreases as the largest clusters increase in size due to absorbing their smaller neighbors. The links connecting them also decrease, with some processes joining together to form thick, straightened bundles [109]. Other weaker processes are pruned to fine-tune the network's wiring [110]. These changes are consistent with the observations for our cell culture system—an initial increase in total process length resulting in the maximal complexity state at around 7 DIV followed by progression towards an optimized state at closer to 17 DIV (Fig 4e–4g, 4j and 4k). Along with pruning, bundling contributes to the observed decreases in $N_{Si}$ and $N_{CNT}$ because the algorithm typically counts each bundle as one link between clusters. Although intriguing, we stress that these shared characteristics are not sufficient to identify our 17DIV network as a small-world system: future experiments would need to establish further structural and transport properties (see Conclusions). We therefore label these networks as cluster networks to emphasize their relatively large clusters.

Neuronal networks have previously been manipulated using patterned substrates to guide cell attachment. Based on previous studies [111–114], we can expect that the neurons in our system utilized neuron-substrate forces to migrate across our smooth $SiO_2$ gaps with average speeds of 10–20 µm/h, covering distances as far as hundreds of microns in the first few weeks of culture [96, 115–118]. Neurons that initially landed in the proximity of our electrode branches had a high chance of their growing processes finding the electrode edges during the first few hours of culturing. The strong cell-VACNT adhesion forces experienced by these neurons would have competed with the neuron-neuron aggregation forces, presumably slowing down cluster formation and resulting in the emergence of the boundary regions (note that these VACNT forces were not sufficiently strong to stop cluster formation completely—as indicated by our observation of mainly small to medium-size clusters on the electrode surfaces and $N_{CNT}$ exhibiting a rise and fall in complexity with culture time—see Fig 4k). Neurons that landed further away from the electrodes would have been less likely to encounter their edges and would therefore have experienced fewer anchor points, mainly in the form of other cells or rough impurities on the surface. In these regions, the neurons therefore had a higher tendency for aggregation and to follow the cluster network formation described above. The desert regions were likely caused by neurons anchored to the VACNT surfaces secreting chemical signals, regulating ion fluxes, neurotransmitters, and specialized signaling molecules [119] which encouraged stronger interaction between neurons on the VACNTs and in the nearby gaps. This process could have left the gaps furthest away from electrodes almost devoid of neurons.

These developments would have been accompanied and supported by an interplay with glial cells. Glia likely started proliferating through cell division and growth and, in this process, acted as a support system for the neurons, following chemical cues [119] that increased their surface coverage close to the neuron-rich regions [120, 121]. This emergence of glial coverage was then likely to support not only neuronal survival and process development, but also migration along their fibers [122, 123] towards the electrodes. Our observations agree with other

studies on smooth surfaces showing that in glial-neuronal co-cultures, glia direct neurons to glial-rich regions using chemical cues [124]. As a sign of their subtle growth interaction, we observed frequent cases of neuronal process development on top of regions covered with glia (S1e Fig).

## Fractal herding behavior

We now consider how the fractal properties of the electrodes influence these inter-dependent networks of glia and neurons. Although previous studies have modelled individual cell loco-motion through environments with complicated geometries [114, 125, 126], the above picture emphasizes the additional roles played by cell growth and assembly behavior (e.g., glial cell division and neuron process bundling and pruning) when considering cells connected in networks. Accordingly, we introduce geometric parameters that relate these behaviors to the fractal geometry of their environment. The fractal electrode design integrates two sets of related, multi-scaled patterns—the branches and the gaps—and our data show that both impact the cell organization favorably. The repeating patterns of the branches build long edges that interface with the gaps. In addition to length, their interpenetrating nature is further amplified by their meandering character. These two branch characteristics were quantified by their total edge length $E_n$ (normalized to the pattern's overall width $W$–Fig 10a-3) and tortuosity $T$, which we measured as the average ratio of branch path length to the direct distance connecting the path's two endpoints (Fig 10b). The gaps between the electrode branches were quantified by their proximities to the branches and their sizes. To measure proximity $P$, the reciprocal of the distance between each gap pixel and its nearest branch pixel was calculated (Fig 10c) and then averaged across all pixel locations in the gap (Methods). The proximity heat maps are displayed in Fig 7a–7c. To quantify the gap sizes, we calculated $A_{min}$ (the smallest rectangular area in the gaps, highlighted by the filled red boxes in Fig 7a–7c), $A_{max}$ (the largest rectangular area, highlighted by the bounded red areas in Fig 7a–7c), and the ratio $A_r$ of $A_{max}$ to $A_{min}$ which quantifies the scaling of the gaps. Finally, $A_c$ (the maximum connected area of the gaps, shown as light gray in Fig 7a–7c) was also calculated. We emphasize the mathematical inter-dependence of $E_n$, $T$, $P$, $A_r$, and $A_c$ established by the fractal geometry and Fig 7d–7h plots their dependences on the electrodes' $D$ and $m$.

Increasing $D$ reduces the rate of pattern shrinkage between levels and so generates larger $E_n$ values, as does increasing the number of levels $m$ (Fig 7d). This large $E_n$ combines with the accompanying increase in $T$ (Fig 7e) to generate the well-defined increase in $P$ observed in Fig 7f. The consequence for neurons and glia is that the large $E_n$, $T$, and $P$ values generated by high $D$ and $m$ patterns offers increased accessibility of electrode edges to the gaps, so increasing favorable interactions between cells in both regions. More specifically, the large edge lengths are likely to accommodate the neurons' tendency to grow processes along the top and bottom edges of the VACNT sidewalls, and for the sidewalls to act as anchor points for neuron clusters in the gaps to adhere to. These effects can explain the increased density of processes connecting the neural networks on the electrodes to those in the gaps observed in Fig 3f. Furthermore, the large average proximity values generated by electrodes with high $E_n$ and $T$ offer shorter distances for cells in the gaps to find and attach to the VACNT edges. This proximity also ensures closeness of the neuron-rich electrodes to the glial cells in the gaps. This is crucial because neurons and glia thrive when in such close proximity [127].

Turning to the parameters describing the gap sizes and how they impact cell behavior, $A_r$ and $A_c$ characterize the 'openness' of the gap patterns for glial cells to cover before being blocked by intruding electrode branches. $A_r$ characterizes the fractal gap sizes that connect together to establish $A_c$. A larger $A_r$ value for an electrode represents a bigger reduction rate in

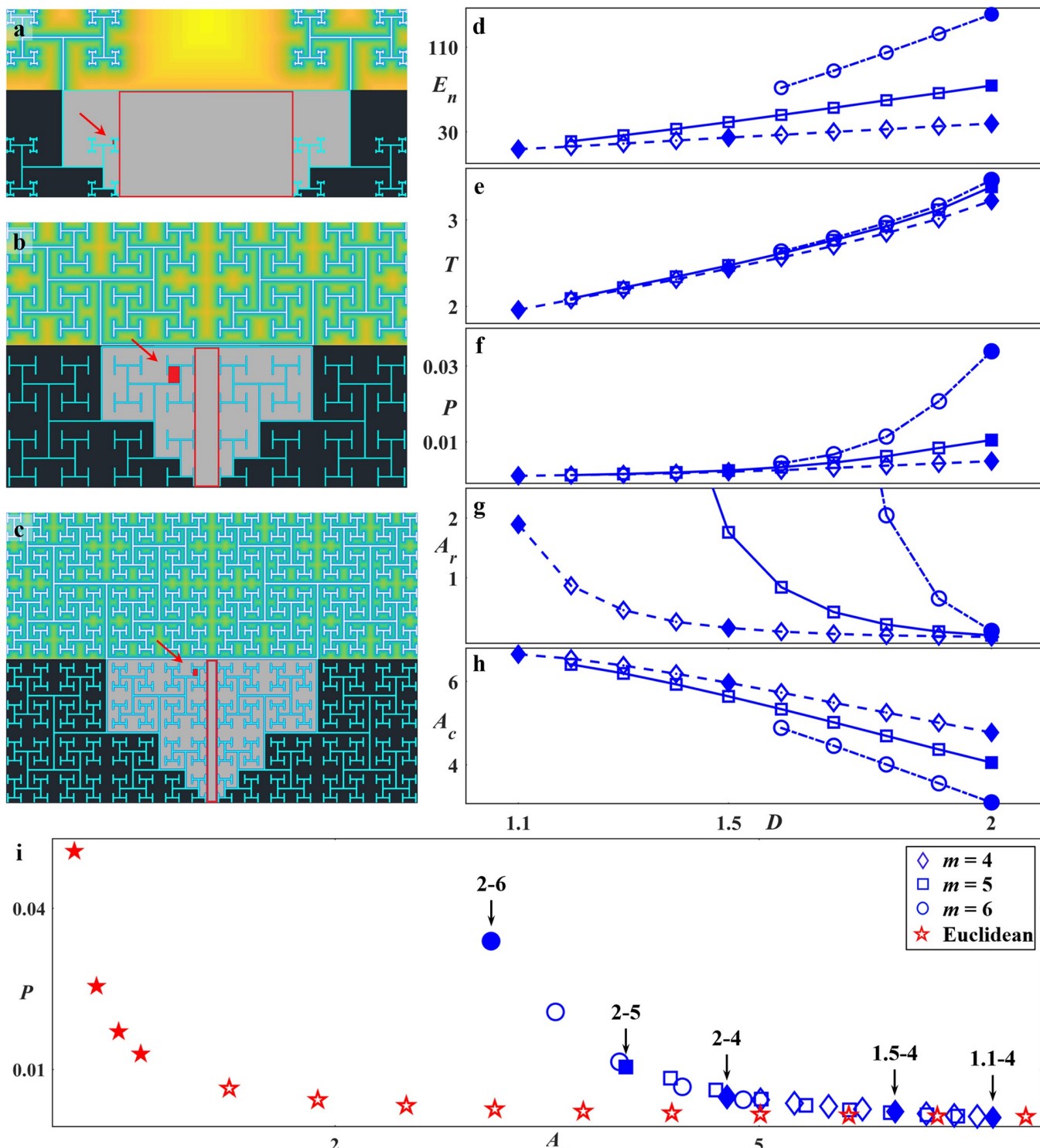

**Fig 7. Schematics of the fractal electrodes and plots of their parameters versus *D* and *m*.** The schematics shown in (a), (b), and (c) are for 1.1–4, 2–4, and 2–5 fractals, respectively. In each case, the top half represents the proximity heat map and the bottom half indicates the largest (bounded red boxes) and smallest (red rectangles indicated by the red arrows) gap areas, along with the largest connected gap area (light gray). (d) Normalized edge length $E_n$, (e) mean tortuosity $T$, (f) mean proximity $P$, (g) Area ratio $A_r$ (y axis multiplied by $10^5$), (h) connected area $A_c$ (y axis multiplied by $10^6$), each plotted vs $D$ for 4, 5, and 6 repeating levels fractals, and (i) $P$ plotted vs $A_c$ for Euclidean electrodes with different $W_{Si}$ values and fractal electrodes with different $D$ and $m$ values (x axis multiplied by $10^6$). In each case, the filled symbols represent electrodes studied experimentally.

gap size between the first and final repeating levels of the fractal ($A_{max}$ vs $A_{min}$), so linking larger regions (that provide the freedom for growth and/or proliferation through cell division) to much smaller regions (that provide close proximity to the neuron-rich electrodes). Electrodes with larger $A_r$ values will also provide vast connected areas $A_c$. The large $A_r$ and $A_c$ values provided by electrodes with small $D$ or $m$ offer physical freedom for helping glial coverage.

Fractals with high $D$ and $m$ provide large $E_n$, $T$, and $P$ but low $A_r$ and $A_c$. These characteristics are predicted to encourage boundary regions and reduce desert regions since there are no large gaps far away from branches. In contrast, fractals with low $D$ and $m$ generate low $E_n$, $T$, and $P$ values but high $A_r$ and $A_c$. These fractals are expected to minimize boundary regions and their vast empty gaps will encourage deserts. Forming between the boundary and desert regions, we expect the growth of cluster regions to be encouraged for mid $D$ and $m$ electrodes. Taken together, these effects suggest that fractals with mid to high $D$ with 4 to 5 repeating levels will promote the most favorable cell interactions. They will enhance glial coverage inside the multi-scaled gaps without restricting the glia and will also prevent the creation of deserts. These glia will contribute to fuel the formation of the neural networks in the cluster regions. The large VACNT edge length of these fractals in close proximity to the $SiO_2$ gaps will enhance growth of neuronal processes in the boundary region connecting the cluster neural networks to those on the VACNT branches.

Within this model, the predicted advantage of H-Tree fractals over Euclidean rows lies in the fact that they provide a connected electrode with an abundance of edges for neurons to follow while allowing glial cells to cover the nearby interconnected, multi-scaled gaps. For example, Fig 7i presents the balance of electrode proximity with gap openness by plotting $P$ versus $A_c$ for the H-Trees and Euclidean rows. The high $P$ and low $A_c$ values of the Euclidean rows studied in our experiments are predicted to be dominated by boundary behavior. However, if Euclidean rows with larger gaps had been fabricated to match the fractal $A_c$ values, their low $P$ values are predicted to be dominated by deserts.

## Analysis

The above predictions of cell-electrode interactions agree with the qualitative observations of Figs 2–4. To understand how they impact the quantitative results in Figs 4–6, we acknowledge that although $N$ and $G$ are useful for summarizing the differences in herding powers for the fractal and Euclidean groups, we need to examine the relationship between their component parameters ($N_{CNT}$, $N_{Si}$, $G_{CNT}$, and $G_{Si}$) in detail. We first examine $G_{Si}$ and the associated glial formations in the gaps by returning to Fig 5e, which plots the median values of $G_{Si}$ against $D$. As expected, the 1.1–4 electrodes supported low $G_{Si}$ values because, although their high $A_r$ and $A_c$ values provided the potential for glial coverage, their drastically reduced $E_n$, $T$, and $P$ values diminished their interaction with the neuron-rich electrode branches, leading to the formation of large deserts typically devoid of glia. In contrast, the 2–4 electrodes promoted larger interaction with the neuron-rich electrodes because of their large $E_n$, $T$, and $P$ values, so leading to their higher observed $G_{Si}$ values when compared to the 1.1–4 electrodes. Because the 2–4 electrodes provided less freedom for glial coverage in the gaps due to their lower $A_r$ and $A_c$, they are expected to have lower $G_{Si}$ values when compared to the 1.5–4 electrodes. In particular, based on the discussions in the last section, the 1.5–4 electrodes should maximize glial coverage by balancing the two competing factors of proximity and freedom. In reality, we note that the observed difference between the 1.5–4 and 2–4 electrodes was not statistically significant, indicating that both achieved this optimization. Compared to these positive performances, the reduction in $A_r$ and $A_c$ with increased repeating levels triggered a reduction in $G_{Si}$ for the 2–6 electrode, a trend completed by envisioning the hypothetical case of a $D = 2$ fractal with

infinite number of repeating levels for which all gaps have vanished and therefore $G_{Si}$ is inevitably zero.

Considering the Euclidean plot of Fig 5a, $G_{Si}$ did not show a statistically significant increase with gap width up to the largest investigated gap of $W_{Si} = 100$ μm suggesting that, although they likely benefitted from large proximities to the neuron-rich electrodes, their gap sizes were insufficient to offer the necessary freedom to encourage large glial coverage. We plot Fig 8 to re-emphasize that the fractal gaps start at the scales of the Euclidean designs and then repeat at increasingly larger sizes. The advantage of connecting to larger gaps for the 1.5–4, 2–4, and 2–5 fractals is demonstrated by plotting $G_{Si}$ versus the minimum gap width $W_{Si-min}$ (the $W_{Si-min}$ values are presented in Table 2). The $G_{Si}$ values for these fractals are notably higher than the Euclidean electrodes with similar $W_{Si-min}$ values (in particular, the 2–5 and 1.5–4 $G_{Si}$ values are significantly higher than the 50 and 100 μm width Euclidean electrodes, respectively, with p = 0.038 and p = 0.021). In contrast, the 2–6 electrodes hold no advantage over the Euclidean rows because their $W_{Si-min}$ gaps were not connected to gaps sufficiently large for proliferation. They have lost the interconnected, multi-scaled freedom of the 1.5–4, 2–4 and 2–5 fractals (as quantified by their lower $A_c$ and $A_r$ values, respectively) and instead approached the more filled character of the rectangular shapes (i.e. higher branch areas, $A_{CNT}$, and lower gap areas, $A_{Si}$. Methods; Table 2). The 1.1–4 electrodes hold no advantage because their $W_{Si-min}$ gaps connected to vast regions dominated by deserts. This behavior is supported by the qualitative inspections of the images shown in Fig 8. In summary, the 1.1–4 fractals were too 'open' and the 2–6 fractal and the Euclidean row geometries too 'restricted'. In this spectrum, the 1.5–4, 2–4, and 2–5 electrodes appear to have the optimal balance for glial coverage provided through integration of a fractal distribution of small and large interconnected areas, interpenetrated by high proximity fractal branches.

To quantify the impact of this glial cell behavior on neuronal process length in the gaps, $N_{Si}$ was plotted as a function of $G_{Si}$ in the scatterplot of Fig 9a. The low $G$ regime and high $G$ regime fractal electrodes are marked with different symbols to detect any possible variations in trends for $N_{Si}$ or $G_{Si}$ between the two regimes. Although the low regime fractals were limited to lower $G_{Si}$, some of the high regime fractals also appeared in this low range, indicating that low $G_{CNT}$ values also played a role in achieving the high $G$ herding powers of the high regime (we will return to $G_{CNT}$ at the end of the Discussion Section). The similarity of these two regimes in terms of their $N_{Si}$ versus $G_{Si}$ relationship indicates a common cell behavior in the gaps, which we will now explain in terms of an interplay of the boundary, cluster, and desert regions.

We begin with the Euclidean electrode data which lies on the left side of the graph, as expected from their relatively low $G_{Si}$ values revealed in Fig 8. These electrodes showed a dominance of boundary regions due to a lack of large gaps to support deserts and a reduction of the cluster networks because of small glial coverage. Cluster networks were nevertheless evident for some Euclidean electrodes with $W_{Si}$ down to 50 μm. This was indicated by the significant decrease in $N_{Si}$ (p = 0.018) in Fig 5c potentially due to increased pruning and bundling as the edges became less influential as the gap widths increased from 50 μm to 100 μm. Nevertheless, the Euclidean electrodes were in general boundary-dominated with large $N_{Si}$ values and the observed sharp increase in $N_{Si}$ with $G_{Si}$ (Fig 9a) was likely due to the supporting role of glial cells on neuronal survival and function. The slight dampening of this rise seen at higher $G_{Si}$ might again be due to the presence of pruning and bundling of the cluster network.

In comparison, the fractals featured significantly fewer neuronal processes (p ≤ 0.00001) in the gaps as quantified by the lower $N_{Si}$ values across the full range of $G_{Si}$ (the few fractal data points residing in the Euclidean data group are the 2–6 fractals, which we have already pointed out have collapsed into the Euclidean condition in terms of their gap behavior). To determine

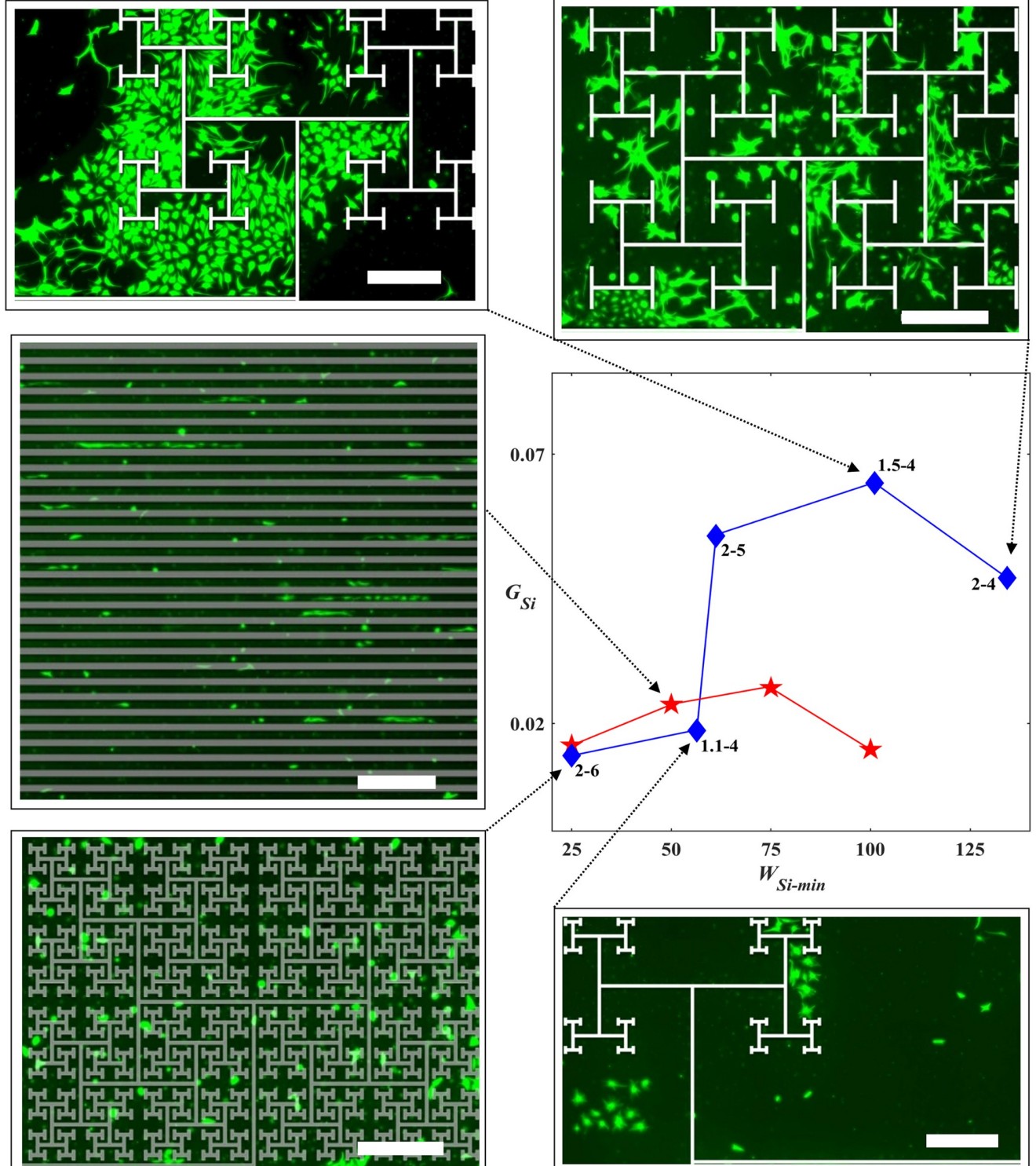

**Fig 8. Glial behavior on SiO₂ surfaces.** Fluorescence images of glial cells (green) at 17 DIV are shown for zoom-in sections showing 1/4th of the full electrode images for the 1.1–4 (bottom right), 1.5–4 (top left), 2–4 (top right), and 2–6 (bottom left) fractals along with the S50C50 (middle left) Euclidean electrodes. White or gray masks are imposed on to the fluorescence images to indicate the locations of the electrodes. Scale bars are 500 μm. A plot of $G_{Si}$ median change against $W_{Si-min}$ is also shown. The dashed arrows connect the images to their corresponding datapoints in the plot. The blue diamond symbols represent fractals. The red pentagrams represent the 17 DIV Euclidean electrodes grouped based on their $W_{Si}$. The error bars correspond to the 95% confidence intervals and are excluded for visual simplicity but range from $\pm 8 \times 10^{-3}$ (for $W_{Si-min} = 100$ μm) to $\pm 4 \times 10^{-2}$ (for 1.5–4). The significance results (see text) are also excluded for clarity.

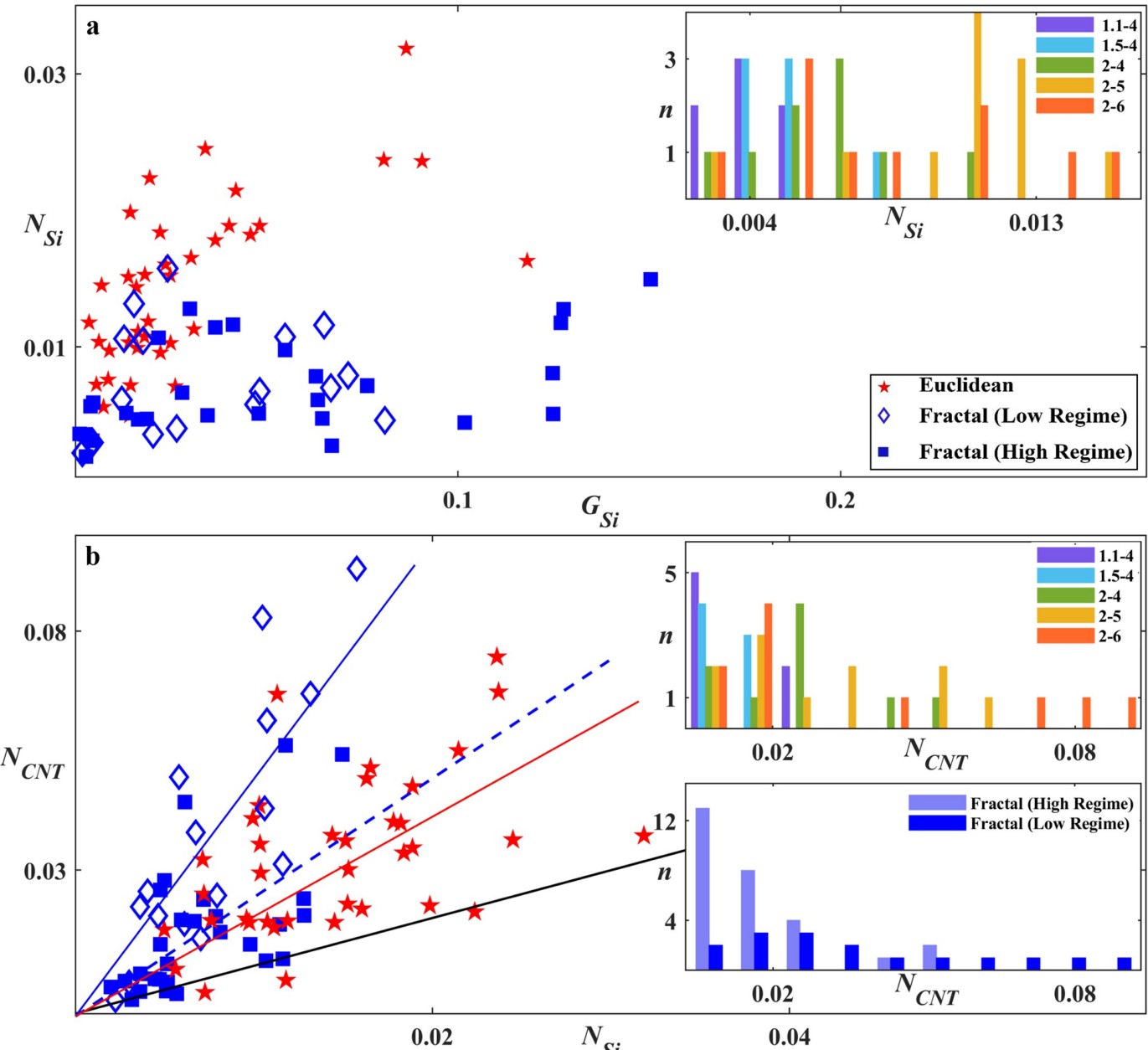

**Fig 9. Study of the relationship between $G_{Si}$, $N_{Si}$, and $N_{CNT}$ for fractal and Euclidean electrodes.** (a) Scatterplot of $N_{Si}$ versus $G_{Si}$ for 17 DIV Euclidean (red pentagram), low (diamond) and high (filled square) regime fractals. Inset of (a) Histogram of the number of electrodes $n$ with a given $N_{Si}$ value for all fractals, grouped according to $D$ and $m$. (b) Scatterplot of $N_{CNT}$ vs $N_{Si}$ for 17 DIV Euclidean, low, and high regime fractals. The solid black line represents the $N_{CNT} = N_{Si}$ condition. The solid blue, dashed blue, and solid red lines are fits through zero for the low regime fractal, high regime fractal, and Euclidean electrodes, respectively. Top inset of (b) Histogram of the number of electrodes $n$ with a given $N_{CNT}$ for all fractals, grouped according to $D$ and $m$. Bottom inset of (b) Histogram of the number of electrodes $n$ with a given $N_{CNT}$ for low and high regime fractals.

the origin of this suppression in $N_{Si}$, the histogram of the number of electrodes $n$ with a given $N_{Si}$ for the various fractal groups is plotted in the Fig 9a inset (the equivalent box plot is shown in S3a Fig). In general, we see that increases in $D$ and $m$ produced higher $N_{Si}$ values. For example, most of the 1.1–4 fractals are located on the plot's left side because their vast deserts dominated over the boundary and cluster regions and supported few processes. Next are the 1.5–4

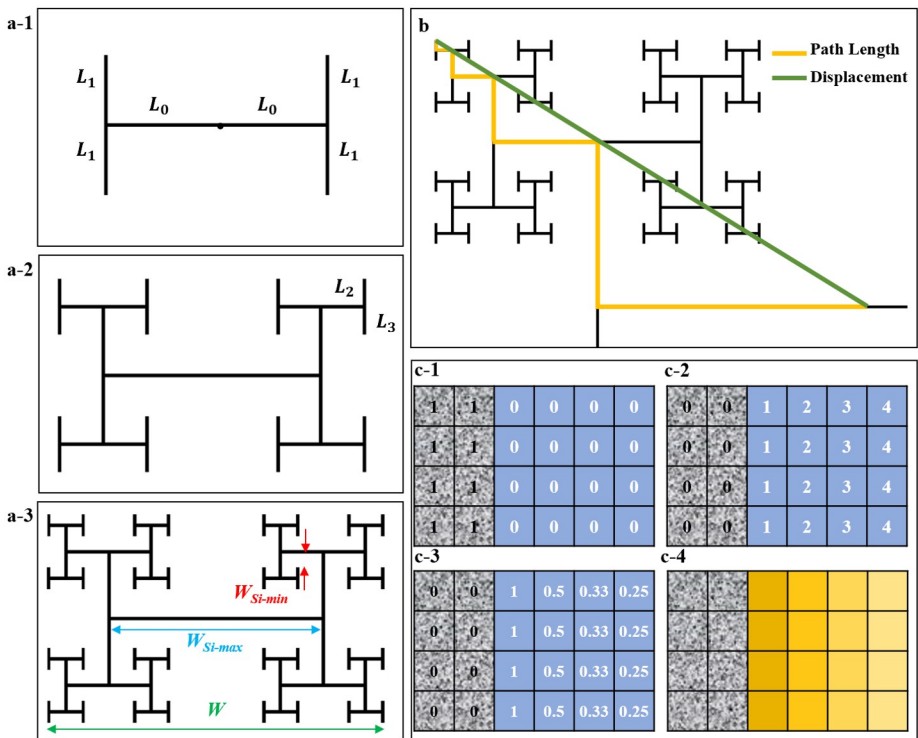

**Fig 10. Schematics of various H-tree parameters.** (a-1 to a-3) Schematics of consecutive stages in the generation of a $D = 1.5$ H-Tree featuring $m = 3$ repeating levels of the H pattern. In panel a-3, the pattern's $W_{Si-min}$, $W_{Si-max}$, and total width $W$ are marked. By incorporating branch segments ranging from $L_0$ to $L_N$, we generated trees with $m$ repeating levels of the H pattern (such that $N = 2m-1$). (b) Schematic representation of the tortuosity calculation for an H-Tree. The yellow line is the path length from the H-Tree center to the end of the final repeating level and is the same for all endpoints. The green line is the displacement from the H-Tree's center to the endpoint of fine-scale branch at the end of the yellow path length (for clarity, not all path lengths and displacements are shown). (c) Schematic demonstration of the conversion of the binary mask to the proximity heat map. (c-1) Matrix representation of the binary mask. The grey and blue pixels represent the VACNT and $SiO_2$ surfaces, respectively. (c-2) Each pixel value is substituted with the minimum distance to the branch pixel. (c-3) Gap pixel values in (c-2) are replaced by their inverse values. (c-4) Colors in the schematic heat map represent closeness to the nearest branch pixels.

fractals because their large cluster networks efficiently bundled and pruned processes. The increasing role of the process-rich boundary regions explains the increase in $N_{Si}$ when moving from the 2–4 to 2–5 fractals. This dependence of $N_{Si}$ on $D$ and $m$ is re-iterated in Fig 5g. (We speculate that the 2–6 fractals have lower $N_{Si}$ values than the 2–5 fractals because the lower $G_{Si}$ and higher $P$ values of the 2–6 fractals increase the tendency of the processes to grow on the VACNTs and their edges rather than the gaps.) Due to the subtle nature of the $N_{Si}$ suppression process, along with the data scatter it generates, we have avoided fitting any form to the $N_{Si}$ versus $G_{Si}$ trend for both the Euclidean and fractal groups.

Equipped with this picture of how $G_{Si}$ and $N_{Si}$ describe cell behavior in the gaps, we now move on to their interaction with the electrode branches. The scatterplot of Fig 9b illustrates how $N_{Si}$ varies with $N_{CNT}$ for the Euclidean, low regime, and high regime fractal electrodes. The solid black line represents successful herding (n.b. almost all electrodes reside above the line, indicating $N_{CNT}$ is larger than $N_{Si}$). We note that $N_{CNT}$ increases with $N_{Si}$ for all three groups, as indicated by their fit lines (note that this line serves simply as a guide to the eye rather than suggesting a strictly linear relationship). A comparison of Fig 9a inset with Fig 9b top inset (and the equivalent box plots in S3b Fig) highlights this trend for the fractal

electrodes: the increase in $N_{CNT}$ values with the rise in $D$ and $m$ has a similar trend to that of the $N_{Si}$ increase. Fractals with a large number of neuronal processes in the gaps and a large electrode interface generated large $N_{CNT}$ values. This is also revealed in Fig 5g and 5h.

Although the fractal parameters influence $N_{CNT}$, they are not sufficiently powerful to produce a statistically significant difference between the low regime fractal and Euclidean groups, a result that highlights the strong adhesive properties of the VACNT surfaces (this is also true of other geometric factors, as indicated by the lack of $N_{CNT}$ dependence on the electrode width for the Euclidean electrodes shown in Fig 5d). For similar $N_{CNT}$ values on the Euclidean and fractal electrodes, the higher Euclidean $N_{Si}$ values (Fig 9a) led to the observed drop in the data gradient for the Euclidean (red slope in Fig 9b) compared to the fractal electrodes (blue slope). Crucially, although the Euclidean and fractal electrodes supported similar numbers of processes, $N_{CNT}$ does not reflect their advantageous locations on the fractal electrodes. A large density of neurons were located at electrode branch edges (e.g., see Fig 2c and 2d) close to the glial coverage in the gaps which ensures neuronal health [20, 21]. The longer edge lengths of the fractals therefore promoted this potential health advantage.

Finally, the collapse in the fractal gradient ($N_{CNT}$ vs $N_{Si}$) when moving from the low to high regime was caused by a drop in $N_{CNT}$ (lower inset of Fig 9b). This may have been induced by a change in glial behavior between the two regimes. In particular, in addition to some fractals supporting $G_{Si}$ values larger than those reached in the low regime (see earlier), $G_{CNT}$ dropped when moving to the high regime. The $G_{CNT}$ values of the Euclidean electrodes were significantly higher than those of the high regime fractals. Although the Euclidean electrodes' more expansive $A_{CNT}$ values (Tables 1 and 2) might have increased the number of seeded glial cells landing on the electrodes, we emphasize that there was no statistically significant difference in $G_{CNT}$ with respect to $A_{CNT}$ for any electrode groups. In contrast, if we consider $W_{CNT}$ dependence, the increase in $G_{CNT}$ between the 20 μm wide fractals and the 100 μm wide Euclidian rows was statistically significant with p = 0.031. It is therefore more likely that the larger widths of the Euclidean electrodes were less restrictive for the subsequent glial growth and that this was responsible for their high $G_{CNT}$ values. The majority of the fractals (61%) lie in the high regime because, along with their higher $G_{Si}$ values, their narrow electrode widths generated small $G_{CNT}$ values. However, there isn't a clear geometric dependence in terms of which fractal electrodes have high and low $G$ values due to 'natural' (i.e. not originating from differences in fabrication and/or culture batches, etc.) statistical variations in the $G_{Si}$ and $G_{CNT}$ values.

When moving from the low to high regime, the increase in the $G_{Si}$ median value was not statistically significant whereas the drop in $G_{CNT}$ was (p ≤ 0.0001). A possible scenario therefore is that the significant drop in $N_{CNT}$ (p = 0.007) is being driven by $G_{CNT}$ (S5c and S5d Fig). The electrode's material properties that caused neurons to thrive needed to be supplemented by chemical cues provided by a small number of glial cells on the VACNT surface. The collapse observed in Figs 6 and 9b was then triggered when this number fell below a critical value (we note that the $G_{CNT}$ variations are responsible for data scatter in S5 Fig). Electrodes with longer edges that promoted stronger process interactions with the boundary regions might be expected to be more robust in terms of preventing this collapse. Accordingly, in general we found that the fractals that were less prone to collapse were those with high $D$ and $m$ values.

## Conclusions

Artificial interfaces that are chemically and physically compatible with biological systems hold great promise for fundamental and applied research and could lead to medical advances that trigger huge impacts across society. Because of their role as the body's main electrical conduction system, neurons have been a major focus of this research field along with glial cells that

serve as the neurons' life-support system. In this study, we exploited variations in surface topography by fabricating textured VACNT and smooth $SiO_2$ regions and highlighted the positive response of neuron-glia co-cultures to this material system for both Euclidean and fractal electrodes. We showed the electrode-cell interactions to be far more subtle than the term herding would suggest. In particular, 'cluster' neural networks in the gap regions near to the electrodes qualitatively displayed some of the structural characteristics of small-world networks: the neurons clustered into large groups of somas supported by glial coverage and connected to other clusters via bundled processes. This network connected to neurons on the electrodes through a 'boundary' region featuring a large density of smaller neuron clusters and processes. For the fractal geometry, we found that electrodes characterized by mid-to-high $D$ and $m$ values optimized the cell response by inducing large glial coverage in the gaps which fueled the cluster neural network. The large branch-gap interface then facilitated the connection of this network to a high density of neuronal processes on the electrode surface. To help clarify this picture, we introduced several parameters–$E_n$, $T$, $P$, $A_r$, and $A_c$– to describe the formation of these networks and how their relative sizes and contributions are determined by the electrode's $D$ and $m$ values (due to the fractal-generated dependencies mapped out in Fig 7d–7h, the dependences of $N_{CNT}$, $N_{Si}$, $G_{CNT}$, and $G_{Si}$ on these parameters followed those expected from their $D$ and $m$ dependencies shown in Fig 5e–5h).

Future studies will define the cell characteristics of the boundary, cluster, and desert regions more precisely to allow their areas, their locations and therefore their contributions to herding to be quantified. This includes analyzing the neural network topography (such as clustering coefficient and shortest pathlengths) of the cluster regions to potentially confirm their small-world characteristics [105, 107]. For the purposes of the current study, we quantified neuronal process length and glial coverage for our morphological measures of the cells. Future studies will consider refined morphological characteristics using more specific cell markers and additional measures such as Sholl analysis. This will allow us to distinguish subcategories of cells, for example, different glial cell types (Müller cells/astrocytes and microglia) [128] and different states of glial activation as well as different neuronal subpopulations such as bipolar and ganglion cells [129]. This will help quantify differences in morphology between the VACNT, boundary, cluster, and desert regions, and so allow them to be differentiated more accurately. Given the different functional roles of, for example, Müller cells/astrocytes and microglia, these distinguishing markers will also allow a greater understanding of the impact of electrode geometry on glial cell activation.

We hope that our approach of studying the interaction of these regions can be used to inform other electrode sizes and designs, along with other material systems and chemical treatments. For example, whereas here we considered Euclidean rows that match the smallest scales of the fractal designs, future experiments could consider wider gaps. Our model predicts that the single-scaled (i.e., $A_r = 1$), minimal tortuosity character of the rectangular rows will not encourage much glial coverage when compared to the fractals. For example, we know that the 2–6 fractals have low $G_{Si}$ values due to the restricted character of their gaps. Therefore, Euclidean rows with larger gaps would need to balance $P$ and $A_c$ more effectively than these 2–6 fractals to encourage glial coverage. Yet, Fig 7i shows that the $P$ values for Euclidean rows with these large $A_c$ values have dropped below those of the desert-inducing 1.1–4 fractals. This might indicate that the Euclidean rows have a narrower parameter range for inducing small-world behavior than the fractals. Although speculative, we note that the median of $G_{Si}$ value of the $W_{Si} = 100$ μm rows is lower than their 75 μm counterparts in Fig 5a perhaps due to the sharp decrease in $P$ observed at the low $A_c$ values in Fig 7i.

For research aimed at refining fractal electrodes to further enhance cell-electrode interactions, we emphasize the insights gained by combining qualitative and quantitative

observations rather than relying on just one approach. These insights highlight the subtleties of herding. For example, the 1.1–4 electrodes in our study achieved favorable neuron herding. However, this was primarily due to a drop in $N_{Si}$ induced by the formation of deserts in the gaps—which are an inherently negative characteristic. Similarly, we note that high regime fractals achieved favorable glial herding through a drop in $G_{CNT}$. However, if $G_{CNT}$ falls below a critical value this can trigger a dramatic decline in $N_{CNT}$ values—which again is inherently negative. We also point out that in our experiments the $G$ threshold of Fig 6 was marked by two characteristics—the Euclidean electrodes failed to surpass this threshold and fractals surpassing the threshold experienced the $N_{CNT}$ collapse. In retrospect, the coincidence of these two characteristics is not inevitable. If we had fabricated Euclidean electrodes with smaller $W_{CNT}$, the associated drop in $G_{CNT}$ would have shifted their $G$ values beyond the dashed line. Similarly, if we had fabricated fractals with larger $W_{CNT}$, more electrodes would have resided below the threshold and avoided the collapse. Finally, in our current study we assumed that greater glial coverage in the gaps is always favorable. However, there might be an optimal coverage above which the positive consequences are less substantial.

Whereas here we focused on the physical arrangements of the cells, future studies targeting applications will analyze the adhesive strength of the neural networks along with their electrical properties. Detection of cell-electrode anchor points by immunostaining (e.g. vinculin and focal adhesion kinase) [130, 131] will provide both a better understanding of the process of network formation and a robust assessment of its attachment to the electrodes. In terms of the electrical properties, we plan to use calcium imaging and microelectrode array (MEA) systems to confirm neuronal stimulation. Such studies will help clarify the impact that the clustering and bundling observed for our neural networks has on the synaptic connectivity efficiency associated with small-world networks [103, 104]. The current *in vitro* studies represent a simple, controlled model for *in vivo* behavior [132]. If these effects are shown to extend to *in vivo* retina-implant applications, they will ensure that a larger number of neuronal processes reside within the stimulating fields generated when the electrode branches are electrically biased. The stimulated processes will connect to small-world neural networks in the gaps, improving the electrode's ability to stimulate the surrounding retinal neurons more efficiently. Furthermore, in contrast to implants that use anti-inflammatory drugs to inhibit glial scarring on their electrode [133] glial cells will be confined to the gaps between our electrodes where their proximity to the neurons will ensure neurons' health, prolonging the stability and functionality of the retina-implant interface.

Finally, in this study we deliberately employed large scale fractal electrodes to manipulate networks of cells. Practically, these large sizes are more applicable to brain stimulation techniques [11] than to retinal implants which require photodiodes featuring 20 μm electrodes [61, 62]. In future studies, we plan to reduce the electrode sizes to see if we can manipulate individual neurons. Although our H-Tree electrodes influenced herding, their shapes are radically different to those of individual neurons (in particular, their straight lines and 90˚ turns are strikingly unnatural). To manipulate individual neurons, we anticipate matching the electrode branches to the precise fractal characteristics of the neuron branches they interact with. In contrast to the exact fractals of Fig 1, neurons belong to a fractal family known as statistical fractals in which random variations prevent exact repetition and only statistical characteristics repeat at different scales. Furthermore, the neurons' fractal characteristics are not primarily driven by the length distributions of the branches (as is the case for the H-Trees) but instead by the way in which the branches 'weave' through space (i.e. their tortuosity) [43]. We anticipate that neuron herding will be increased by matching these 'bio-inspired' electrodes to the fractal weave properties of the neurons. Additionally, our studies of interactions between natural and artificial fractal systems might have broader implications, for example for

neuromorphic computing. Whereas neuromorphic circuits typically incorporate components that mimic neuronal behavior, for example resistive switches that correspond to synaptic connections [134], our circuits mimic the network architecture of the neurons.

## Methods

### Euclidean patterns

We fabricated 104 Euclidean electrodes all with the same overall pattern width $W = 6000$ μm. One group features rows of VACNT forests of constant width $W_{CNT} = 100$ μm separated by $SiO_2$ rows with widths that vary between electrode patterns in 25 μm increments from 25 to 100 μm. For the second group, the widths of the VACNT and $SiO_2$ rows are equal and vary in 25 μm increments from 25 to 100 μm between electrode patterns (Table 1).

In addition, we generated simulated Euclidean patterns with $W = 6000$ μm, $W_{CNT} = 100$ μm, and $W_{Si}$ ranging from 200 to 1100 μm increasing in 100 μm increments. We calculated proximity $P$ and maximum connected area $A_c$ for these Euclidean electrodes. To quantify $P$, the distance between each gap pixel and its nearest branch pixel was calculated. The proximity matrix was then created by assigning the equivalent 1/(minimum distance) value to each gap pixel (Fig 10c). $P$ was then calculated by averaging over all matrix elements in the gaps. $A_c$ was calculated by multiplying $W$ with $W_{Si}$ for each pattern. The 1100 μm upper gap limit was chosen such that its $A_c$ approximately matched the largest $A_c$ belonging to the 1.1–4 fractal pattern.

### Fractal pattern generation and quantification

An example of the H-Tree fractal generation process is shown in Fig 10a. The largest of the repeating H patterns is constructed from 2 horizontal branch segments (each a straight line of length $L_0$) and 4 perpendicular branch segments (length $L_1$), where:

$$L_1 = \frac{L_0}{2^{\frac{1}{D}}} \tag{1}$$

The fractal dimension $D$ then sets how this H pattern scales with each repetition such that the $n^{\text{th}}$ branch segment's length is given by:

$$L_n = \frac{L_0}{2^{\frac{n}{D}}} \tag{2}$$

We fabricated 44 H-Trees with branches of constant width $W_{CNT} = 20$ μm for $D = 1.1–4$, 1.5–4, and 2, each with $m = 4$ repeating levels (Fig 1). The overall width of these patterns was chosen to be $W = 6020$ μm to match the width of the Euclidean patterns ($L_0$ was adjusted to ensure a constant $W$ for the various $D$ values). We also fabricated $D = 2$ H-Trees with 5 and 6 repeating levels (Fig 1). For these H-Trees, $W = 6262$ μm was chosen to ensure that the smallest distance between branches ($W_{Si-min}$ in Fig 10a-3) was 25 μm for the $m = 6$ H-Tree, so matching the narrowest Euclidean gap. The $W_{Si-min}$ values for each of the fractal electrodes are shown in Table 2. For the simulated H-Trees, the pattern's overall width was set to $W = 6262$ μm and $D$ was varied from 1.1 to 2 in 0.1 increments. When calculating their geometric parameters (listed below), we excluded the $D$ values and repeating levels that generated overlapping branches.

The branches were quantified by their lengths and by the degree to which they meandered across the surface. The total edge length $E$ was given by:

$$E(N, W, D, W_{CNT}) = \frac{2A_{CNT}}{W_{CNT}} + W_{CNT} \times 2^{(N+1)} - \sum_{n=1}^{N} W_{CNT} \times 2^n \tag{3}$$

where $A_{CNT}$ is the surface area covered by all of the branches in the H-Tree:

$$A_{CNT}(N, W, D, W_{CNT}) = W_{CNT}L_b - \sum_{n=1}^{N} 2^n (W_{CNT}^2/2) \tag{4}$$

and $L_b$ is the total length of the branches:

$$L_b(N, W, D, W_{CNT}) = \left(\frac{W - W_{CNT}}{2\sum_{n=0}^{(N-1)/2} 1/2^{\frac{2n}{D}}}\right) \sum_{n=0}^{N} 2^{n+1}/2^{\frac{n}{D}} \tag{5}$$

To allow a comparison of different sized H-Trees, the normalized edge length $E_n$ was then defined as:

$$E_n = {}^E/_W \tag{6}$$

We calculated the tortuosity $T$ to quantify the branch meandering. For each branch, the path length from the H-Tree's center to the endpoint of each fine-scale branch was divided by the equivalent displacement. These tortuosity values were then averaged across all possible endpoints in the H-Tree (Fig 10b). These calculations of $E_n$ and $T$ were used to plot their $D$ dependences in Fig 7d and 7e, respectively.

The gaps between the branches were quantified by their proximities to the branches and by their sizes. To quantify the mean proximity $P$, the distance between each gap pixel and its nearest branch pixel was calculated. The proximity matrix was then created by assigning the equivalent 1/(minimum distance) value to each gap pixel (Fig 10c). $P$ was then calculated by averaging over all matrix elements in the gaps. This matrix was used to generate the heat maps in Fig 7a–7c, and the $P$ vs $D$ plots in Fig 7f.

To quantify the sizes of the H-Tree's gaps, we considered the following three surface areas: $A_{min}$ is the smallest rectangular area in the gaps (i.e. the filled red boxes in Fig 7a–7c identified by the arrows) while $A_{max}$ is the largest rectangular area (i.e. the bounded red areas in Fig 7a–7c). These two areas are defined such that they are confined by the branches without any intersection or interruption. $A$ is the maximum connected area of the gaps between the branches in the H-Tree (shown as the light gray areas in Fig 7a–7c). $A_{min}$, $A_{max}$, and $A_c$ are given by:

$$
\begin{aligned}
&A_{min}(N, W, D, W_{CNT}) \\
&= \left(\frac{W - W_{CNT}}{2\sum_{n=0}^{(N-1)/2} 1/2^{\frac{2n}{D}}}\right)^2 \left(1/2^{\frac{N-2}{D}} \times 1/2^{\frac{N-1}{D}}\right) - W_{CNT}\left(\frac{W - W_{CNT}}{2\sum_{n=0}^{(N-1)/2} 1/2^{\frac{2n}{D}}}\right)\left(1/2^{\frac{N-2}{D}} + 1/2^{\frac{N-1}{D}}\right) + W_{CNT}^2 \tag{7}
\end{aligned}
$$

$$
\begin{aligned}
A_{max}(N, W, D, W_{CNT}) =\ & 2\left(\frac{W - W_{CNT}}{2\sum_{n=0}^{(N-1)/2} 1/2^{\frac{2n}{D}}}\right)^2 \left(1 - \sum_{n=1}^{(N-1)/2} 1/2^{\frac{2n}{D}}\right)\left(\sum_{n=0}^{(N-1)/2} 1/2^{\frac{2n+1}{D}}\right) \\
& - \left(\frac{W - W_{CNT}}{2\sum_{n=0}^{(N-1)/2} 1/2^{\frac{2n}{D}}}\right) W_{CNT}\left(1 + \sum_{n=0}^{(N-1)/2} 1/2^{\frac{2n+1}{D}} - \sum_{n=1}^{(N-1)/2} 1/2^{\frac{2n}{D}}\right) + (W_{CNT}/2)^2
\end{aligned} \tag{8}
$$

$$A_c(N, W, D, W_{CNT})$$

$$= 2\left(\frac{W - W_{CNT}}{2\sum_{n=0}^{(N-1)/2} 1/2^{\frac{2n}{D}}}\right)^2 \left(\sum_{n=0}^{(N-1)/2} 1/2^{\frac{2n+1}{D}} - \sum_{n=1}^{(N-1)/2} \sum_{m=n}^{(N-1)/2} 1/2^{\frac{2m+2n+1}{D}}\right)$$

$$- \left(\frac{W - W_{CNT}}{2\sum_{n=0}^{(N-1)/2} 1/2^{\frac{2n}{D}}}\right) W_{CNT} \left(\sum_{n=1}^{(N-1)/2} \sum_{m=2n+1}^{N} 2^m/2^{2n+\frac{m}{D}} + \sum_{n=0}^{N} 1/2^{\frac{2n}{D}}\right) \quad (9)$$

To quantify the scaling of the H-Tree gaps, the ratio of the largest to smallest areas $A_r = A_{max}/A_{min}$ was plotted versus $D$ for the $m$ = 4, 5, and 6 H-Trees in Fig 7g. To facilitate comparisons of different sized H-Trees, $A_c$ was plotted vs $D$ in Fig 7h. Tables 1 and 2 summarize some geometric measurements for the Euclidean and fractal electrodes: branch width ($W_{CNT}$), Euclidean gap width ($W_{Si}$), minimum H-Tree gap width ($W_{Si-min}$), maximum H-Tree gap width ($W_{Si-max}$), overall pattern width ($W$), total surface area of the branches ($A_{CNT}$), total gap surface area ($A_{Si}$), and the electrode bounding area ($A_{bounding}$).

## Carbon nanotube synthesis and characterization

The VACNTs were synthesized on silicon wafers with a 300 nm thermal oxide ($SiO_2$) top layers following procedures that were described elsewhere [83]. The whole 2-inch wafers were cleaned, dehydrated, and exposed to hexamethyldisilazane (HMDS, Sigma Aldrich, St Louis, MO) for 20 minutes. A spin-coated photoresist layer was patterned using photolithography techniques. The whole wafer contained 16 individual electrode patterns of various types. A 2–5 nm Al adhesive layer was then deposited thermally followed by a 3–5 nm Fe catalyst layer. The photoresist layer was then lifted off using acetone accompanied by sonication for 30 seconds, so leaving the patterned Al-Fe layer. The wafer was then cut into individual patterns. The VACNTs were then grown on these catalyst patterns in a 2-inch quartz tube using a 2:1 mixture of ethylene:hydrogen (200 and 100 SCCM, respectively) for 3 minutes at 650°C. A 600 SCCM flow of Argon kept the tube clean. This technique resulted in patterned electrodes consisting of entangled 'forests' of VACNTs covering a $6 \times 4$ mm$^2$ region of an approximately 1 cm$^2$ wafer (Fig 1 and Fig 11). The electrodes were then stored in integrated circuit (IC) trays in a desiccator cabinet. The topographical characteristics of the top and sidewalls of the VACNTs, their heights, and general conditions of patterned wafers were inspected using a ZEISS-Ultra-55 scanning electron microscope (SEM). No visual differences were observed between samples belonging to different geometries and fabrication runs. VACNT heights were in the range 20–45 μm. The wafers patterned with the VACNT electrodes were placed in 4-well culture plates (Sarstedt, Newton, NC), one wafer per well.

## Dissociated retinal cell cultures

*Wildtype C57BL/6 mice* were kept at animal welfare services at University of Oregon. Handling and all other procedures involving the mice were performed according to protocols approved by the University of Oregon's Institutional Animal Care and Use Committee under protocol 16–04, in compliance with National Institutes of Health guidelines for the care and use of experimental animals. The animals had full time access to fresh water and food supplies. Dissociated retinal cell cultures were grown under protocols described elsewhere [38, 83, 129]. Briefly, postnatal day 4 mice were euthanized by decapitation and retinas quickly dissected and kept in Dulbecco's Modified Eagle Medium (DMEM- ThermoFisher Scientific, Waltham, MA) containing high-glucose, sodium pyruvate, L-glutamine, and phenol red. Four retinas

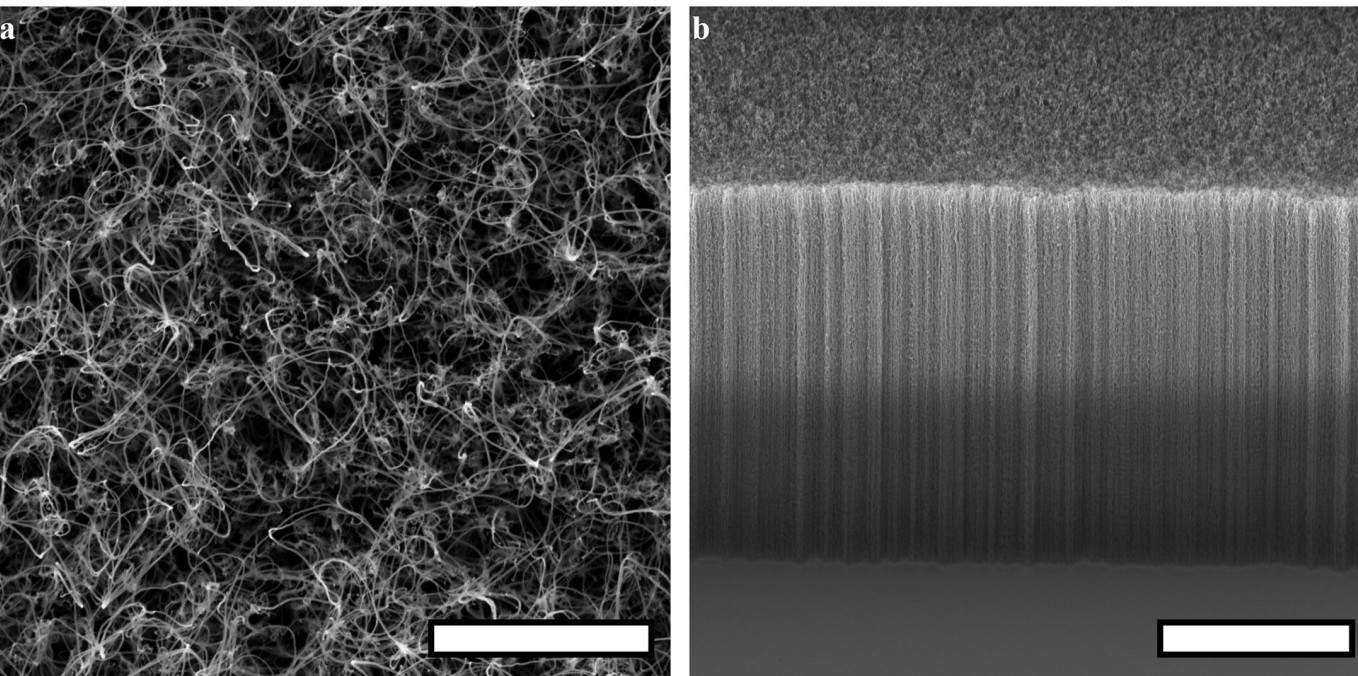

**Fig 11. SEM images of patterned VACNT forests taken before the culturing experiments.** (a) Top-down view of entangled VACNTs on the forest's top surface, (b) View of the sidewall of a VACNT row taken at a 40˚ angle. Scale bars are 2 and 10 μm, respectively.

were transferred into an enzyme solution containing DMEM, papain (Worthington Biochemical Corporation, Lakewood, NJ), and L-cysteine (Sigma-Aldrich, St Louis, MO). The digested retinas were carefully rinsed with DMEM and transferred to DMEM containing B27 (Sigma-Aldrich, St Louis, MO) and L-glutamine-penicillin-streptomycin (Sigma-Aldrich, St Louis, MO). The dissociated retina solution was centrifuged, and the cell pellet was resuspended in the DMEM/B27/antibiotic solution. The cell suspension (500 μL) was subsequently seeded onto each well containing a VACNT electrode. The live cell density as measured by a hemocytometer was $(3.6 \pm 0.5) \times 10^6$ cellsmL$^{-1}$. This cell density was converted to an area density through multiplying by the volume of the cell suspension and dividing by the surface area of a culture well (1.9 cm$^2$). The average area density for the H-Tree cultures was $(9.3 \pm 1.1) \times 10^3$ cellsmm$^{-2}$. As examples, the average number of cells that were typically seeded on the VACNT branches of a 1.1–4 and 2–6 fractal electrode are $7.6 \times 10^3$ and $8.2 \times 10^4$, respectively. Each culture experiment included a minimum of 8 and a maximum of 16 electrodes from diverse geometries.

The cells seeded onto Euclidean electrodes were cultured for 3, 7, and 17 DIV and onto fractal electrodes for 17 DIV at 37˚C and 5% $CO_2$. The culture medium was first changed at 3 DIV and then every other day until the end of the culture time for the 7 and 17 DIV cultures. Some culture experiments included narrower fractal patterns (10 μm wide VACNT branches) as well as 7 DIV fractal electrodes that were beyond the scope of this work and were excluded from the analyses. It is worth noting that no protocols such as precoating the surfaces with poly-D-lysine (PDL) or poly-L-lysine (PLL) were used to increase the neuronal adhesion to the different surface types in these experiments. These protocols were intentionally avoided since it was intended that cells interact with VACNT and $SiO_2$ surfaces with unaltered surface properties.

## Immunocytochemistry

The immunocytochemistry protocol used has been previously described [38, 129]. Briefly, the cells were fixed with paraformaldehyde (PFA), rinsed with a phosphate buffered solution (PBS), and pre-incubated in PBS-complete, containing PBS, Triton-X (Sigma-Aldrich, St Louis, MO), bovine serum albumin (BSA, Sigma-Aldrich, St Louis, MO), goat normal serum, and donkey normal serum (Jackson ImmunoResearch, West Grove, PA). The cells were subsequently incubated with PBS-complete containing the primary antibodies, monoclonal mouse anti-β-tubulin III (concentration: 1:1500; neuronal marker; Antibody ID: AB_1841228; Clone number: 2G10; Sigma-Aldrich, St Louis, MO), and polyclonal rabbit anti-glial fibrillary acidic protein (concentration: 1:1500; GFAP, glial cell marker; Antibody ID: AB_10013382; Catalogue number: Z0334; Agilent, Santa Clara, CA). Next, the cells were rinsed and incubated with PBS-complete containing the secondary antibodies polyclonal Cy3 goat anti-mouse IgG (Concentration: 1:200; Antibody ID: AB_2338680; Product code: 115-165-003; Jackson ImmunoResearch, West Grove, PA) and polyclonal AlexaFluor 488 donkey anti-rabbit IgG (Concentration: 1:400; Antibody ID: AB_2313584; Product code: 711-545-152; Jackson ImmunoResearch, West Grove, PA). After removal of the secondary antibody solution, the cells were rinsed and the wafers transferred to microscope slides and mounted with Vectashield containing DAPI (fluorescent cell nuclear marker that binds to DNA) (Vector Laboratories, Burlingame, CA).

## Fluorescence microscopy

A Leica DMi8 inverted fluorescence microscope was used to take 20× images in the Cy3 (excited at 550 nm, emission peak at 565 nm), AlexaFluor 488 (excited at 493 nm, emission peak at 519 nm), and DAPI (excited at 358 nm, emission peak at 461nm) channels for all electrodes. The top VACNT and bottom $SiO_2$ surfaces were imaged separately with the focus being adjusted to these surfaces. The 2048×2048 pixel$^2$ (662.65×662.65 μm$^2$) FOVs in each channel were then stitched together using an automated stitching algorithm with 10% overlap at the edges of neighboring FOVs to create full electrode images.

## Post-culture SEM imaging

For post-culture SEM imaging, cells were fixed in 1.25% and 2.5% glutaraldehyde solutions in deionized (DI) water for 10 and 20 minutes, respectively. After rinsing 3 times in PBS for 10 minutes each, the wafers were submerged in increasing concentrations of ethanol (50%-100%) for 15 minutes each for dehydration. They were then submerged in a 2:1 solution of ethanol:HMDS for 20 minutes followed by a 20-minute rinse in 1:2 ethanol:HMDS and finally a 20-minute rinse in 99.9% HMDS. The cells were left in fresh 99.9% HMDS overnight to let it evaporate. The electrodes were then coated with a 20 nm thick layer of gold before SEM imaging.

## Neurite length and glial area measurements

Binary masks were created for each Euclidean and fractal electrode based on the width of the rows and branches, $D$, and $m$. These masks were then applied to all acceptable FOVs (those without any abnormalities such as VACNT deformations) within an electrode to distinguish between the VACNT and $SiO_2$ surfaces. An automated image analysis based on a previously reported algorithm [135] was integrated with the binary mask algorithm to detect and measure the total neuronal process length per FOV on the VACNT and $SiO_2$ surfaces separately. This algorithm was insensitive to overlapped neuron processes such as bundles or multiple

processes following the edges of the branches and detected them as one process. This resulted in an undercount of neurons especially on the VACNT surfaces. However, accounting for this undercount in neuronal processes would have further emphasized the favorability of VACNT surfaces for neurons. The total process lengths on the VACNT and $SiO_2$ surfaces and total VACNT and $SiO_2$ areas per electrode sample were calculated by summing these parameters for all FOVs across the electrode. The normalized total process length for VACNT and $SiO_2$ per electrode sample were then calculated as follows:

$$N_{CNT} = \frac{Total\ process\ length\ on\ VACNT\ per\ electrode}{Total\ VACNT\ area\ available\ per\ electrode} \tag{10}$$

$$N_{Si} = \frac{Total\ process\ length\ on\ silicon\ per\ electrode}{Total\ silicon\ area\ available\ per\ electrode} \tag{11}$$

For the glia, a semi-automated thresholding algorithm integrated with the binary mask algorithm was used to detect and measure the total area covered by glial cells on the VACNT and $SiO_2$ surfaces per FOV. The normalized total glial coverage area for the VACNT and $SiO_2$ surfaces were then calculated as follows:

$$G_{CNT} = \frac{Total\ glial\ coverage\ area\ on\ VACNT\ per\ electrode}{Total\ VACNT\ area\ available\ per\ electrode} \tag{12}$$

$$G_{Si} = \frac{Total\ glial\ coverage\ area\ on\ silicon\ per\ electrode}{Total\ silicon\ area\ available\ per\ electrode} \tag{13}$$

To reduce the error in detecting neuron processes and glial areas around the edges of the electrodes on the VACNT and $SiO_2$ surfaces, FOVs were inspected and the sizes of the masks were adjusted manually if necessary to allow for correct detection of in-focus features on either of the surfaces. In order to quantitatively compare the total neuronal process lengths and total glial coverage areas for the VACNT and $SiO_2$ surfaces, three 'herding' parameters were introduced. Neuron herding $N$, glia herding $G$, and combined herding $GN$ were defined as follows:

$$N = \frac{N_{CNT}}{N_{Si} + N_{CNT}} \tag{14}$$

$$G = \frac{G_{Si}}{G_{CNT} + G_{Si}} \tag{15}$$

$$GN = G \times N \tag{16}$$

$N$ and $G$ values greater than 0.5 indicate successful guiding of neuronal processes and glial cells to the desired VACNT or $SiO_2$ surfaces, respectively. Specifically, the $N > 0.5$ condition corresponds to more neuronal processes existing on the VACNT surfaces than the $SiO_2$ gaps. The $G > 0.5$ condition corresponds to more glial coverage in the $SiO_2$ gaps than on the VACNT surfaces. $GN$ was calculated to compare combined herding powers between various electrode groups.

## Statistical analysis

The nonparametric Kruskal-Wallis test for significance followed by the nonparametric post-hoc Dunn's test were used in MATLAB (R2019b) to compare the medians of neuronal and

**Table 3. Total number of each electrode geometry used in these experiments as well as number of independent cultures including each electrode design.**

| Culture time | 3 DIV | | 7 DIV | | 17 DIV | |
|---|---|---|---|---|---|---|
| Samples | number of electrodes | number of independent cultures | number of electrodes | number of independent cultures | number of electrodes | number of independent cultures |
| Euclidean electrodes | | | | | | |
| S100C100 | 6 | 3 | 11 | 7 | 8 | 6 |
| S75C100 | 3 | 2 | 4 | 4 | 6 | 4 |
| S50C100 | 3 | 1 | 5 | 4 | 6 | 4 |
| S25C100 | 3 | 1 | 5 | 4 | 4 | 2 |
| S75C75 | 3 | 1 | 6 | 3 | 4 | 3 |
| S50C50 | 3 | 1 | 6 | 3 | 5 | 3 |
| S25C25 | 2 | 1 | 6 | 3 | 5 | 2 |
| Fractal electrodes | | | | | | |
| 1.1–4 | | | | | 7 | 2 |
| 1.5–4 | | | | | 7 | 5 |
| 2–4 | | | | | 9 | 4 |
| 2–5 | | | | | 11 | 5 |
| 2–6 | | | | | 10 | 4 |

glial parameters among different groups against various null hypotheses (e.g., the fractal electrodes were tested against the null hypothesis that $D$ and $m$ would not impact $G_{Si}$) with the significance set at $p < 0.05$. A total number of 104 Euclidean electrodes from 13 independent cultures and 44 fractal electrodes from 8 different cultures were used in the experiments (Table 3). Mixtures of electrode geometries were included in each independent culture.

## Supporting information

**S1 Fig. Examples of SEM and fluorescence images of neurons and glia on VACNT and SiO$_2$ surfaces.** (a) SEM image showing a neuronal process bridging a 50 μm gap between two VACNT rows (7 DIV). (b) SEM image showing neuronal processes on the top surface of a VACNT electrode (7 DIV). (c) SEM image of a neuronal process on the top surface of a VACNT electrode (17 DIV). (d) Merged fluorescence image of a region on a 2–5 fractal showing β-tubulin III labelled neuronal processes (red) attached to and following VACNT branches and GFAP labelled glial cells (green) in the SiO$_2$ gaps (17 DIV). (e) SEM image of a glial cell and neuronal processes on the smooth SiO$_2$ surface (17 DIV). Cell bodies and processes are false-colored in (a), (b), and (e). Scale bars are 10 μm in (a) and (e), 2 μm in (b) and (c), and 75 μm in (d).
(TIF)

**S2 Fig. Study of the relationship between (a) $G_{Si}$, (b) $G_{CNT}$, (c) $N_{Si}$, and (d) $N_{CNT}$ with the SiO$_2$ to VACNT area ratio for 17 DIV Euclidean electrodes.** No statistical significance was detected between any pair for all glial and neuronal parameters.
(TIF)

**S3 Fig. Study of the neuronal behavior on SiO$_2$ and VACNT surfaces for fractal electrodes.** Statistical analysis showing boxplots for $N_{Si}$ (a) and $N_{CNT}$ (b). Stars in (a) indicate the degree of significance: * denotes $p \leq 0.05$ and *** denotes $p \leq 0.001$. The red plusses in panel (a) are outliers. No significance was observed in $N_{CNT}$.
(TIF)

**S4 Fig. Comparison of glial and neuronal behavior on VACNT and SiO$_2$ surfaces for the Euclidean (17 DIV) and fractal electrodes.** Statistical analysis showing boxplots of $G_{Si}$ vs $G_{CNT}$ for (a) 17 DIV Euclidean and (c) fractals. As well as $N_{Si}$ vs $N_{CNT}$ for (b) 17 DIV Euclidean and (d). Stars in all panels indicate the degree of significance: **** denotes p $\leq$ 0.0001. The red plusses are outliers.
(TIF)

**S5 Fig. Comparison of the low and high regime fractal electrode groups.** Plots of $G_{Si}$ (a) and $N_{CNT}$ (b) against $G_{CNT}$ showing the different cell behaviors on the VACNT and SiO$_2$ surfaces for the low and high regime fractals. Statistical analysis showing boxplots for $G_{CNT}$ (c) and $N_{CNT}$ (d). No significance was observed in $N_{Si}$ and $G_{Si}$ between the 2 groups. Stars in (c) and (d) indicate the degree of significance: ** denotes p $\leq$ 0.01 and **** denotes p $\leq$ 0.0001. The red plusses in panels (c) and (d) for the high regime fractals are outliers.
(TIF)

**S1 Table. Median values of all glial and neuronal parameters at 3, 7, and 17 DIV for Euclidean and at 17 DIV for fractal electrode types.**
(DOCX)

# Acknowledgments

RPT is a Cottrell Scholar of the Research Council for Science Advancement. We thank M. Pluth (University of Oregon) for providing the opportunity and training for the fluorescence microscopy imaging system. We thank W. Griffiths (University of Oregon) for his contribution in the discussion of the results and K. Zappitelli (University of Oregon) for her contributions to experimental design, the development of the VACNT synthesis process, cell culturing, immunocytochemistry, and microscopy imaging.

# Author Contributions

**Conceptualization:** Saba Moslehi, Conor Rowland, Julian H. Smith, William J. Watterson, David Miller, Cristopher M. Niell, Benjamín J. Alemán, Maria-Thereza Perez, Richard P. Taylor.

**Data curation:** Saba Moslehi, Conor Rowland, Julian H. Smith, William J. Watterson.

**Formal analysis:** Saba Moslehi, Conor Rowland, Julian H. Smith.

**Funding acquisition:** William J. Watterson, Cristopher M. Niell, Benjamín J. Alemán, Maria-Thereza Perez, Richard P. Taylor.

**Investigation:** Saba Moslehi, Conor Rowland, Julian H. Smith, William J. Watterson, David Miller.

**Methodology:** Saba Moslehi, Conor Rowland, Julian H. Smith, William J. Watterson, David Miller, Cristopher M. Niell, Benjamín J. Alemán, Maria-Thereza Perez, Richard P. Taylor.

**Project administration:** Richard P. Taylor.

**Resources:** Cristopher M. Niell, Benjamín J. Alemán, Maria-Thereza Perez, Richard P. Taylor.

**Software:** Saba Moslehi, Conor Rowland, Julian H. Smith, William J. Watterson.

**Supervision:** Cristopher M. Niell, Benjamín J. Alemán, Maria-Thereza Perez, Richard P. Taylor.

**Validation:** Saba Moslehi, Conor Rowland, Julian H. Smith, Richard P. Taylor.

**Visualization:** Saba Moslehi, Conor Rowland, Julian H. Smith, Maria-Thereza Perez, Richard P. Taylor.

**Writing – original draft:** Saba Moslehi, Richard P. Taylor.

**Writing – review & editing:** Saba Moslehi, Conor Rowland, Julian H. Smith, William J. Watterson, David Miller, Cristopher M. Niell, Benjamín J. Alemán, Maria-Thereza Perez, Richard P. Taylor.

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
