## [Decision Letter · Decision Letter 0]

20 Jan 2022

PONE-D-21-36805Controlled Assembly of Retinal Cells on Fractal and Euclidean ElectrodesPLOS ONE

Dear Dr. Moslehi,

Thank you for submitting your manuscript to PLOS ONE. After careful consideration, we feel that it has merit but does not fully meet PLOS ONE’s publication criteria as it currently stands. Therefore, we invite you to submit a revised version of the manuscript that addresses the points raised during the review process.

Overall, the external reviewers agree of the importance of your manuscript and were impressed with the extensive dataset you included. Reviewer #1 furthemore proposed potential additional analysis, namely detection of anchor points through a focal adhesion analysis and calcium imaging. You may want to consider to complement your dataset with this analysis. Alternatively please clarify your decision to not include such analysis in your response and in the manuscript. Some minor revisions were also proposed, e.g. statistical significance should be included in Fig 3 as suggested by Reviewer #2. Please submit your revised manuscript by Mar 06 2022 11:59PM. If you will need more time than this to complete your revisions, please reply to this message or contact the journal office at plosone@plos.org. Please include the following items when submitting your revised manuscript:A rebuttal letter that responds to each point raised by the academic editor and reviewer(s). You should upload this letter as a separate file labeled 'Response to Reviewers'.A marked-up copy of your manuscript that highlights changes made to the original version. You should upload this as a separate file labeled 'Revised Manuscript with Track Changes'.An unmarked version of your revised paper without tracked changes. You should upload this as a separate file labeled 'Manuscript'.

We look forward to receiving your revised manuscript.

Kind regards,

Maria Asplund, PhD

Academic Editor

PLOS ONE

Journal Requirements:

“All authors participated in the study design; SM, CR, JHS, and WJW created the neuron and glial analysis algorithms; DM and BJA built the CVD system; DM, BJA, SM, and WJW developed the VACNT synthesis process; BJA conceived and implemented the VACNT platform; SM and CR fabricated VACNT electrodes, MTP developed the cell culturing and immunochemistry protocols; 46 SM, CR, JHS, and WJW performed retinal cell cultures and fluorescence microscopy; SM, CR, and JHS performed image processing and statistical data analysis; SM, CR, JHS, and RPT performed data analysis and validation; SM, CR, JHS, and RPT designed the figures; SM, CR, JHS, MTP, and RPT helped edit figures; SM and RPT drafted the manuscript; RPT coordinated the project; RPT, MTP, BJA, and CMN acquired the funding; all authors edited the manuscript.”

“RPT:

•            W. M. Keck Foundation

http://www.wmkeck.org/

•            The Living Legacy Foundation

https://www.thellf.org/

•            The Ciminelli Foundation

•            University of Oregon

https://www.uoregon.edu/

MTP

•            The Swedish Research Council - # 2016-03757

https://www.vr.se/english.html

•            NanoLund at Lund University

https://www.nano.lu.se/

•            Stiftelsen för Synskadade i f.d. Malmöhus Län

https://synskadademalmohus.se/

•            Crown Princess Margareta’s Committee for the Blind

Reviewers' comments:

Reviewer's Responses to Questions

**Comments to the Author**

1. Is the manuscript technically sound, and do the data support the conclusions?

Reviewer #1: Yes

Reviewer #2: Yes

2. Has the statistical analysis been performed appropriately and rigorously? 

Reviewer #1: Yes

Reviewer #2: Yes

3. Have the authors made all data underlying the findings in their manuscript fully available?

Reviewer #1: Yes

Reviewer #2: Yes

4. Is the manuscript presented in an intelligible fashion and written in standard English?

Reviewer #1: Yes

Reviewer #2: Yes

5. Review Comments to the Author

Reviewer #1: The manuscript entitled „Controlled Assembly of Retinal Cells on Fractal and Euclidean Electrodes” presents an extensive study on how the geometry of electrodes affects the behaviour of retinal cells and glial cells on the surface: their attachment, preferred anchorage places, surface density, process lengths and other parameters that are derived from fractal-like geometry of neurons.

I am impressed with the extensity of research and the amount of valuable results that were derived from a straightforward experimental design. The rationale of this study has been carefully planned, and the results allowed to confirm the initial hypothesis about the effect of fractal geometry on a cell co-culture. I am sure that this study will be an important contribution to the development of biomedical electrodes, not only restricted to interfacing retinal cells, but also other cells and tissues that can be electrically stimulated.

Before accepting the manuscript for publication, I would like Authors to consider the following aspects:

- The introduction section will benefit if the Authors devote a paragraph to the analysis of a geometry of retinal cells. It would be great to cite the research showing that changes in fractal parameters of retinal cells could be associated with cellular processes, such as neurodegeneration (providing that such studies exist).

- A large part of geometrical analysis is focused on retinal cells. I was wondering if glial cells could be also analysed with the use of additional morphometric descriptors. Maybe it would be worth to investigate whether astrocytes could become activated on the electrodes with different geometries? You may check this paper: https://onlinelibrary.wiley.com/doi/10.1002/glia.22684

Further studies:

- Detection of anchor points through a focal adhesion analysis.

- Calcium imaging to verify how electrical pulses are transferred between neurons through different electrodes.

Reviewer #2: The manuscript by Molehi et al. examines an engineered electrode interface with H-tree fractal vs. Euclidian patterned geometries on the behavior of dissociated retinal cell types. The vertical aligned carbon nanotube (VACNT) surfaces appeared to favor the attachment or localization of neurons, whilst the smooth SiO2 surfaces favored glial coverage. Here the authors are using the material properties and surface interactions to 'herd' cells based on their behavior. The fractal parameter of the patterning was therefore investigated to optimize cell-electrode interactions, which were largely quantified morphologically but not functionally verified (ie, by electrophysiology).

The manuscript is fascinating and novel in both its aims and methods, thus worthy of publishing to reach the wide readership of PLoS. It is very well written, although a few minor improvements are suggested below, which should be made to make the results more clear to the reader.

Minor points:

- Figure 3 j1-4 labels are difficult to read as text is too bold. Perhaps this is only in the PDF render, but one might choose another font or not use bold titles. 'Desert' was particularly hard to make out but only after reading the text was it clear what it could be.

-please mark statistical significance on Fig3 with asterices or similar to help the reader.

6. PLOS authors have the option to publish the peer review history of their article (what does this mean?). If published, this will include your full peer review and any attached files.

Reviewer #1: **Yes: **Katarzyna Krukiewicz

Reviewer #2: No

---

## [Author Response · Author response to Decision Letter 0]

8 Feb 2022

Dear Dr. Asplund,

We thank the reviewers for their positive reviews of our manuscript, “Controlled Assembly of Retinal Cells on Fractal and Euclidean Electrodes” (PONE-D-21-36805). We have addressed all of their points in the revised manuscript and we believe that their suggestions have improved our submission. We have also addressed all the editorial requirements. Below, we list our revisions in detail. In each case, we have left the reviewer and editorial comments in black font and listed our responses in blue font.

Yours sincerely,

Saba Moslehi, PhD

Post-doctoral Research Fellow

University of Oregon - Department of Physics

sabam@uoregon.edu

PONE-D-21-36805 reviewer comments-2/7/22. Review Comments to the Author

Reviewer #1: The manuscript entitled “Controlled Assembly of Retinal Cells on Fractal and Euclidean Electrodes” presents an extensive study on how the geometry of electrodes affects the behaviour of retinal cells and glial cells on the surface: their attachment, preferred anchorage places, surface density, process lengths and other parameters that are derived from fractal-like geometry of neurons.

I am impressed with the extensity of research and the amount of valuable results that were derived from a straightforward experimental design. The rationale of this study has been carefully planned, and the results allowed to confirm the initial hypothesis about the effect of fractal geometry on a cell co-culture. I am sure that this study will be an important contribution to the development of biomedical electrodes, not only restricted to interfacing retinal cells, but also other cells and tissues that can be electrically stimulated.

Before accepting the manuscript for publication, I would like Authors to consider the following aspects:

- The introduction section will benefit if the Authors devote a paragraph to the analysis of a geometry of retinal cells. It would be great to cite the research showing that changes in fractal parameters of retinal cells could be associated with cellular processes, such as neurodegeneration (providing that such studies exist).

 The reviewer raises a crucial point about the importance of fractality and the use of fractal analysis for studying the morphological characteristics of biological systems especially retinal cells. In response, we have expanded the introduction to cite research that focuses on the characterization of different cell types in the nervous system using fractal analysis. We have separated the references into groups of studies focusing on neurons and glial cells. Additionally, we have categorized them based on the aim of the study, whether it was conducted to distinguish between various subcategories or to diagnose pathological conditions. Given the extensive amount of valuable research conducted in this field, we acknowledge that the list of references added are by no means comprehensive and will serve as a guide to the readers of PLOS One who will be interested in the fractality of biological systems. The added paragraph is now included in the Introduction (lines 87-93).

- A large part of geometrical analysis is focused on retinal cells. I was wondering if glial cells could be also analysed with the use of additional morphometric descriptors. Maybe it would be worth to investigate whether astrocytes could become activated on the electrodes with different geometries? You may check this paper: https://onlinelibrary.wiley.com/doi/10.1002/glia.22684

 We appreciate the reviewer’s suggestions for studying the morphological characteristics of glial cells to investigate their level of activity in response to the electrode materials and geometry. Such studies are possible following the previously developed methods such as Sholl analysis and fractal box counting methods (e.g., Kang, K. et al. “The complex morphology of reactive astrocytes controlled by fibroblast growth factor signaling”, Glia (2014), 62(8), 1328-1344 and Behar, T. “Analysis of Fractal Dimension of O2A Glial Cells Differentiating in Vitro”, Methods (2001), 24(4), 331–9). However, because astrocytes and Müller cells both express glial fibrillary acidic protein (GFAP-marked with immunocytochemistry techniques) in our in vitro system, performing the morphology analysis on our current fluorescent images without having the ability to distinguish between the two cell types would complicate interpretations of the results. Because of the importance of such studies for understanding the fundamentals of cell-electrode interactions, we therefore propose future experimental designs that use individual retinal cell markers for all neuron and glial cell types. This will enable us to not only distinguish between the different glial cell types but also between various neurons in our in vitro system. Whereas the current studies focused on neuronal process length and glial covering area as the quantifiable measurements of cell morphology and behavior, future studies will focus on more sophisticated morphological characteristics of all cell types. This would not only enable us to determine the glial cell activity level on the electrodes (for Müller cells, astrocytes, and microglia) but also study the effects of their morphological characteristics on the formation of the three ‘boundary’, ‘cluster’, and ‘desert’ regions on the SiO2 surface. These suggested future studies are outlined in our future goals in the Conclusions (lines 717-727).

Further studies:

- Detection of anchor points through a focal adhesion analysis.

- Calcium imaging to verify how electrical pulses are transferred between neurons through different electrodes.

 We thank the reviewer for suggesting future studies aimed at achieving a better understanding of interactions of cells with the electrode (detection of anchor points) as well as investigating a working electrode (calcium imaging of the electrical activity of the neural network). These suggested future studies are now discussed in the Conclusions (lines 758-766).We propose the use of focal adhesion markers such as vinculin or focal adhesion kinase to detects and analyze the anchor points as well as calcium imaging of the neurons’ electrical activity as part of our future endeavors. These studies will quantify the effects of electrode geometry on the adhesive strength of the neural networks along with their electrical properties. 

Reviewer #2: The manuscript by Molehi et al. examines an engineered electrode interface with H-tree fractal vs. Euclidian patterned geometries on the behavior of dissociated retinal cell types. The vertical aligned carbon nanotube (VACNT) surfaces appeared to favor the attachment or localization of neurons, whilst the smooth SiO2 surfaces favored glial coverage. Here the authors are using the material properties and surface interactions to 'herd' cells based on their behavior. The fractal parameter of the patterning was therefore investigated to optimize cell-electrode interactions, which were largely quantified morphologically but not functionally verified (ie, by electrophysiology).

The manuscript is fascinating and novel in both its aims and methods, thus worthy of publishing to reach the wide readership of PLoS. It is very well written, although a few minor improvements are suggested below, which should be made to make the results more clear to the reader.

Minor points:

- Figure 3 j1-4 labels are difficult to read as text is too bold. Perhaps this is only in the PDF render, but one might choose another font or not use bold titles. 'Desert' was particularly hard to make out but only after reading the text was it clear what it could be.

 We thank the reviewer for their keen observation. All figures have now been edited to meet PLOS One figure requirements and have been tested and approved by the Preflight Analysis and Conversion Engine (PACE) digital diagnostic tool. We hope this resolves the reviewer’s concern on the clarity of the figures.

-please mark statistical significance on Fig3 with asterices or similar to help the reader.

 We thank the reviewer for their helpful suggestion for making our results more accessible to the readers. In particular, we appreciate that statistical significances should be included in the figures whenever possible in order to help summarize the details provided in the text. Accordingly, we have now included the standard star notation representation of statistical significance results in Fig. 4 and Fig. 5 and have edited the figure legends to include the interpretation of the stars.

Editorial requirements:

and

 We thank the editor for pointing out the style requirements for the manuscript submission. The revised manuscript meets all the PLOS One style requirements.

“All authors participated in the study design; SM, CR, JHS, and WJW created the neuron and glial analysis algorithms; DM and BJA built the CVD system; DM, BJA, SM, and WJW developed the VACNT synthesis process; BJA conceived and implemented the VACNT platform; SM and CR fabricated VACNT electrodes, MTP developed the cell culturing and immunochemistry protocols; 46 SM, CR, JHS, WJW, and MTP performed retinal cell cultures and fluorescence microscopy; SM, CR, and JHS performed image processing and statistical data analysis; SM, CR, JHS, and RPT performed data analysis and validation; SM, CR, JHS, and RPT designed the figures; SM, CR, JHS, MTP, and RPT helped edit figures; SM and RPT drafted the manuscript; RPT coordinated the project; RPT, MTP, BJA, and CMN acquired the funding; all authors edited the manuscript.”

Please remove any funding-related text from the manuscript and let us know how you would like to update your Funding Statement. 

 The added funding statement “RPT, MTP, BJA, and CMN acquired the funding.” is now removed from the Contributions statement. We have updated the Funding Statement as follows:

RPT, CMN, and BJA:

• W. M. Keck Foundation

http://www.wmkeck.org/

RPT, WJW, and MTP:

• The Pufendorf Institute

https://www.pi.lu.se/en/pufendorf-ias

RPT:

• The Living Legacy Foundation

https://www.thellf.org/

• The Ciminelli Foundation

• University of Oregon

https://www.uoregon.edu/

MTP:

• The Swedish Research Council - # 2016-03757

https://www.vr.se/english.html

• NanoLund at Lund University

https://www.nano.lu.se/

• Stiftelsen för Synskadade i f.d. Malmöhus Län

https://synskadademalmohus.se/

• Crown Princess Margareta’s Committee for the Blind

 All the data and the related code is now available at 10.6084/m9.figshare.19127894 and we have updated our Data Availability Statement.

 We declare that the reference list is complete and correct and does not include any retracted studies. Upon revising the manuscript the references listed below were added to the list:

44 – 58, 60, 68 – 71, 79, 128, 130, and 131.

---

## [Editor Report · Decision Letter 1]

7 Mar 2022

Controlled assembly of retinal cells on fractal and Euclidean electrodes

PONE-D-21-36805R1

Dear Dr. Moslehi,

We’re pleased to inform you that your manuscript has been judged scientifically suitable for publication and will be formally accepted for publication once it meets all outstanding technical requirements.

Kind regards,

Maria Asplund, PhD

Academic Editor

PLOS ONE
---

## [Editor Report · Acceptance letter]

10 Mar 2022

PONE-D-21-36805R1 

Controlled assembly of retinal cells on fractal and Euclidean electrodes 

Dear Dr. Moslehi:

I'm pleased to inform you that your manuscript has been deemed suitable for publication in PLOS ONE. Congratulations! Your manuscript is now with our production department. 

Kind regards, 

on behalf of

Dr. Maria Asplund 

Academic Editor

PLOS ONE